# Nitrate restricts nodule organogenesis through inhibition of cytokinin biosynthesis in *Lotus japonicus*

Jieshun Lin [1], Yuda Purwana Roswanjaya [2,3], Wouter Kohlen [2], Jens Stougaard [1] & Dugald Reid [1✉]

Legumes balance nitrogen acquisition from soil nitrate with symbiotic nitrogen fixation. Nitrogen fixation requires establishment of a new organ, which is a cytokinin dependent developmental process in the root. We found cytokinin biosynthesis is a central integrator, balancing nitrate signalling with symbiotic acquired nitrogen. Low nitrate conditions provide a permissive state for induction of cytokinin by symbiotic signalling and thus nodule development. In contrast, high nitrate is inhibitory to cytokinin accumulation and nodule establishment in the root zone susceptible to nodule formation. This reduction of symbiotic cytokinin accumulation was further exacerbated in cytokinin biosynthesis mutants, which display hypersensitivity to nitrate inhibition of nodule development, maturation and nitrogen fixation. Consistent with this, cytokinin application rescues nodulation and nitrogen fixation of biosynthesis mutants in a concentration dependent manner. These inhibitory impacts of nitrate on symbiosis occur in a *Nlp1* and *Nlp4* dependent manner and contrast with the positive influence of nitrate on cytokinin biosynthesis that occurs in species that do not form symbiotic root nodules. Altogether this shows that legumes, as exemplified by *Lotus japonicus*, have evolved a different cytokinin response to nitrate compared to non-legumes.

[1] Department of Molecular Biology and Genetics, Aarhus University, Aarhus, Denmark. [2] Laboratory for Molecular Biology, Wageningen University, Wageningen, Netherlands. [3] Centre of Technology for Agricultural Production, Agency for the Assessment and Application of Technology (BPPT), Jakarta, Indonesia. ✉email: dugald@mbg.au.dk

Nitrogen deficiency is the most common nutritional limitation to plant growth. Legumes can overcome this limitation by acquiring nitrogen through the establishment of a symbiotic relationship with nitrogen fixing rhizobia. The establishment of a new organ (a nodule) and transfer of resources to the bacterial partner makes symbiotic nitrogen fixation less favourable than uptake of soil nitrate. Therefore, nitrate is preferentially acquired and nodule development is inhibited in soils with high nitrate levels[1].

Nodule development is initiated through a common symbiotic pathway shared with establishment of arbuscular mycorrhizal fungi inside plant roots[2–4]. A major downstream target in the establishment of nodules is the induction of cytokinin synthesis, transport and signalling[5–10]. Cytokinin is essential for nodule organogenesis, as exemplified by loss of nodule development in cytokinin receptor mutants[5,6,11,12]. Cytokinin biosynthesis during nodule development is controlled partially redundantly, with multiple *Ipt*, *Log* and *Cyp735a* genes contributing to increased cytokinin levels[8,9]. Insertion mutants in *Ljipt4* have minor phenotypes in nitrate-free conditions, while *LjIpt3* knock-down or knock-out has been reported to have variable phenotypes in several studies[8,13,14]. Synthesis of trans-Zeatin by *LjCyp735a* is induced during nodule development, but knock-out mutants do not display a nodulation phenotype in *L. japonicus*[8]. Stimulation of cytokinin signalling through biosynthesis and receptor activation is also sufficient to trigger nodule development, even in the absence of rhizobia[7,8,15,16]. This cytokinin signalling regulates expression of central components of nodulation signalling, including the transcription factor NIN (Nodule Inception)[17] via distal *cis*-elements in the *Nin* promoter[18]. Cytokinin- and NIN-dependent signalling also initiates negative feedback of nodule organogenesis and infection via induction of CLE (CLAVATA3/ESR-related) peptides and the AON (Autoregulation of Nodulation) pathway[19–21] and cytokinin may play further roles in AON in the shoot[14]. Nitrate induction of CLE peptides[22,23] allows the AON pathway to integrate signals from both prior nodulation events and soil nitrate availability to balance nodule development with plant resources.

Nitrate signalling depends on uptake and perception by nitrate transceptors (transporter-receptors) of the NRT1.1 family[24,25]. The majority of transcriptional responses to nitrate are then controlled by the action of NIN-Like proteins (NLPs)[26,27]. In legumes, NLP signalling is tightly integrated with symbiotic signalling with NLPs regulating symbiotic signalling both directly and through competition for NIN binding sites[22,28,29]. Loss-of-function mutants in legume NLPs (e.g. *Ljnrsym1*/*Ljnlp4* and *Mtnlp1*) therefore show nitrate-resistant symbiosis phenotypes. In addition to negative regulation of gene expression by nitrate, NLPs participate in activation of some symbiotic genes[29].

In addition to local responses to nitrate, plants possess systemic regulatory circuits allowing response to nitrate to be coordinated between roots and shoots. Cytokinin signalling is one of the pathways underlying this systemic signalling of nitrate availability[30]. Outside of legumes, induction of cytokinin biosynthesis by nitrate has been described in several species including maize[31], rice[32] and Arabidopsis[33]. Several cytokinin biosynthesis (IPT, LOG and CYP735A enzymes) and transport (ABCG14) components have subsequently been shown to coordinate plant responses to nitrogen. For example, *AtIPT3* and *AtABCG14* activity in the root vasculature can increase cytokinin export in response to nitrate supply[33–35], while *AtIPT3* and *AtCYP735A2* have been implicated in NLP-dependent nitrate signalling[36–38]. In rice, four IPT genes and a resulting accumulation of cytokinin were identified as nitrate and ammonium responsive[32].

Given the positive role of cytokinin in nodule development, in contrast to the inhibitory role of nitrate in this process, it remains to be seen how and whether cytokinin synthesis and signalling play equivalent signalling and coordination roles in nitrogen signalling in legumes. Here, we investigate this link and find nitrate is inhibitory to cytokinin biosynthesis in the model legume *Lotus japonicus*. This provides a regulatory target for nitrate signalling, ensuring nodule development and soil nitrate is balanced and our work implies additional regulatory mechanisms may be recruited or amplified in legumes to signal nitrogen availability.

## Results

**Rhizobia-induced cytokinin biosynthesis is inhibited by nitrate.** In response to rhizobia, *L. japonicus* induces the expression of cytokinin biosynthesis genes, including *Ipt2*; *Log1*; and *Log4*, to trigger nodule organogenesis[8]. This induction occurs primarily in the region of emerging root hairs at the root tip, known as the susceptible zone (Fig. 1A). While environmental nitrate induces cytokinin synthesis in many species, in legumes high nitrate can inhibit nodule organogenesis, raising the question of how legumes deal with this apparent paradox. To investigate the effect of nitrate on cytokinin biosynthesis during nodule initiation, we first analysed the expression of cytokinin biosynthesis genes. In the absence of nitrate, *Ipt2*; *Ipt3*; *Ipt4*; *Log1;Log4* and *Cyp735a* are upregulated in the susceptible zone at one and/or two days after rhizobia inoculation, in line with previous results (Fig. 1A–E; Supplementary Fig. 1)[8]. However, in the presence of high nitrate which is inhibitory to nodule initiation (5 mM $KNO_3$), the relative transcript abundance of *Ipt2*; *Log1* and *Log4* was lower at both one and two days post inoculation (dpi), relative to plants grown in the absence of nitrate (Fig. 1B–D; Supplementary Fig. 1), while *Ipt4* was lower in nitrate conditions at one dpi and *Ipt3* at two dpi (Supplementary Fig. 1). *Cyp735a*, which is required for production of *trans*-Zeatin (*tZ*), did not show a nitrate-dependent reduction (Fig. 1E). In the absence of inoculation, only *Ipt4* and *Cyp735a* (elevated) showed significant differences in transcript levels in the nitrate condition.

To confirm that this nitrate-induced reduction of transcript levels resulted in altered levels of cytokinin, we quantified the cytokinin ribosides and bases in the same tissue and conditions as the gene expression studies two days post inoculation. In agreement with the gene expression analysis, nitrate exposure significantly reduces isopentenyladenine (iP) relative to nitrate-free plants irrespective of inoculation status (Fig. 1F). The levels of *tZ* were not significantly different between the two nitrate conditions (Fig. 1G).

**Cytokinin biosynthesis mutants show reduced iP content in N sufficient conditions.** Given that iP production is inhibited by nitrate, we hypothesised that *ipt* biosynthesis mutants may exacerbate this reduction. We therefore analysed the cytokinin content of *ipt3-1*, *ipt4-1* and the *ipt3-2 ipt4-1* double mutant two days post rhizobia inoculation in the absence and presence of nitrate (Fig. 2). In uninoculated conditions, all of these mutants showed reduced iP content in the susceptible zone relative to the wild-type Gifu. Similar to the wild-type, in all cases iP content increased after inoculation, likely through activity of the symbiotic responsive *LjIpt2*[8] and confirming the redundancy present in the cytokinin biosynthesis pathway. In high nitrate both the *ipt3-1* and *ipt4-1* mutants showed reduced iP content after inoculation relative to plants grown without nitrate. Under high nitrate, the *ipt4-1* and *ipt3-2 ipt4-1* mutants showed the lowest iP content following inoculation with 54.5% and 58.8% of wild-type levels in the same condition (Fig. 2A). While different genotypes

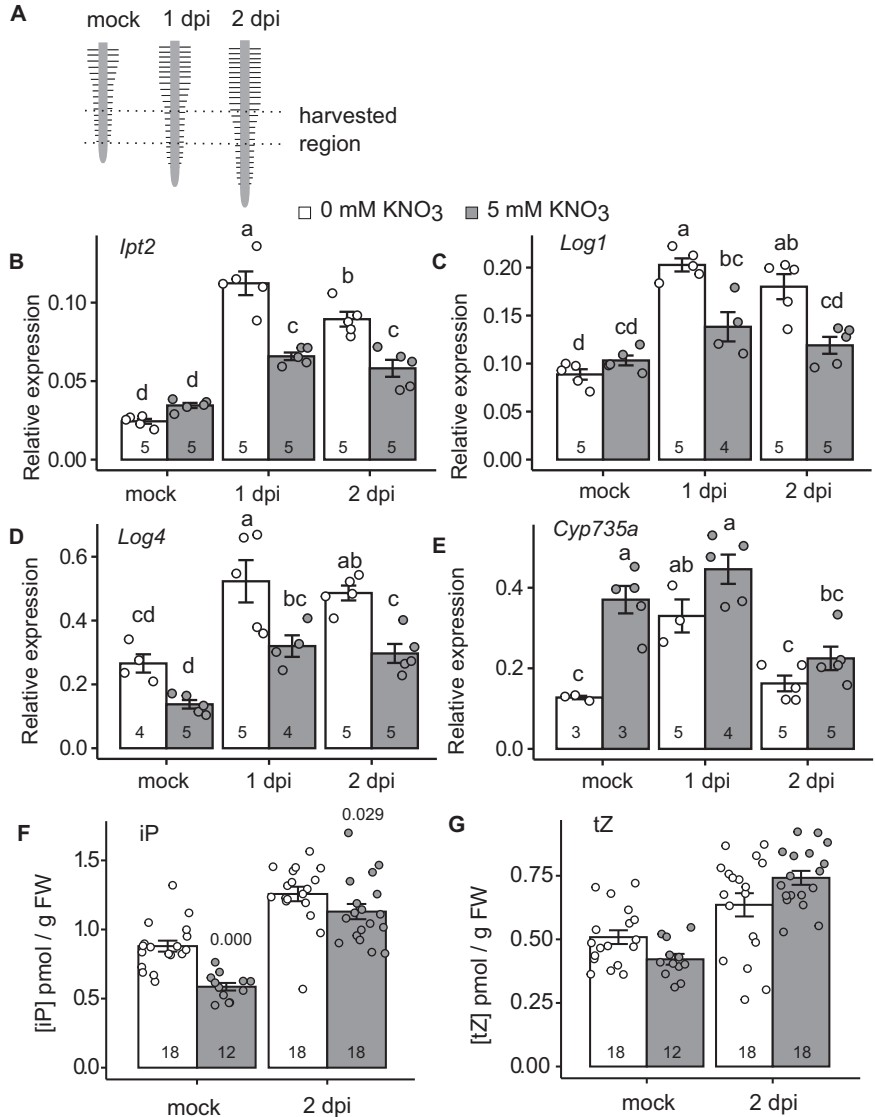

**Fig. 1 Nitrate inhibits induction of symbiotic cytokinin biosynthesis. A** The root zone which was susceptible to nodule initiation at time of inoculation was used for qRT-PCR and cytokinin analysis. **B–E** Relative transcript abundance of selected cytokinin biosynthesis genes by qRT-PCR 1 and 2 days post inoculation (dpi) with *M. loti* in the absence and presence of 5 mM $KNO_3$. Ubiquitin is used as a reference gene. **E, F** Cytokinin-free base content (iP and *tZ*) analysed in mock or 2 days post inoculation with *M. loti* in the absence and presence of 5 mM $KNO_3$. Bars show mean ± SE for the *n* value shown on each bar. Significant differences among different conditions are indicated by letters ($p < 0.05$) as determined by ANOVA and Tukey post-hoc testing for **B–E**. Pairwise *P*-values as determined by Tukey testing are available in the Source Data file. Significant differences between nitrate presence and absence conditions as determined by Wilcoxon rank-sum test is indicated by the annotated *P* values in **F, G**.

showed altered levels of *tZ*, nitrate did not influence the *tZ* levels in any genotype or inoculation condition (Fig. 2B).

**Cytokinin biosynthesis mutants are hyper-sensitive to nitrate inhibition of nodule organogenesis but not infection.** To assess the impact of the reduced iP on nodule development, we investigated the nodulation phenotypes of *ipt3-1*, *ipt4-1* and *ipt3-2 ipt4-1* relative to Gifu in the absence or presence of 5 mM nitrate. In the absence of nitrate, *ipt3-1* and *ipt4-1* mutants did not show significantly reduced nodulation while *ipt3-2 ipt4-1* formed slightly fewer nodules in these conditions (Fig. 3A–D). Note that we were persistently unable to reproduce an increased nodulation phenotype for *Ljipt3* mutants, despite using the same alleles as a previous study[14]. In high nitrate, nodule initiation (indicated here as total nodules) and nodule maturation (as determined by nodules acquiring a distinct pink-red colour characteristic for

fully developed nodules expressing leghemoglobin) are significantly more impaired on *ipt3-1* (maturation), *ipt4-1* (initiation and maturation) and *ipt3-2 ipt4-1* (initiation and maturation) roots relative to wild type (Fig. 3A–D). In particular, on *ipt3-2 ipt4-1* roots, only small bumps that do not fully mature are observed (Fig. 3A).

The impact of impaired nodule maturation on nitrogen fixation was assessed by Acetylene Reduction Assay (ARA). In line with our scoring of nodule development, in the absence of nitrate, there is no significant difference in ARA in these mutants. Under high nitrate conditions, Gifu shows a significant 35.4% reduction in ARA activity relative to nitrate-free conditions (Fig. 3E). We found that *ipt3-1*, *ipt4-1* and *ipt3-2 ipt4-1* mutants all showed a significantly greater sensitivity to nitrate inhibition, exhibiting 80.2, 64.3 and 91.7% reduction in ARA respectively relative to nitrate-free conditions (Fig. 3E).

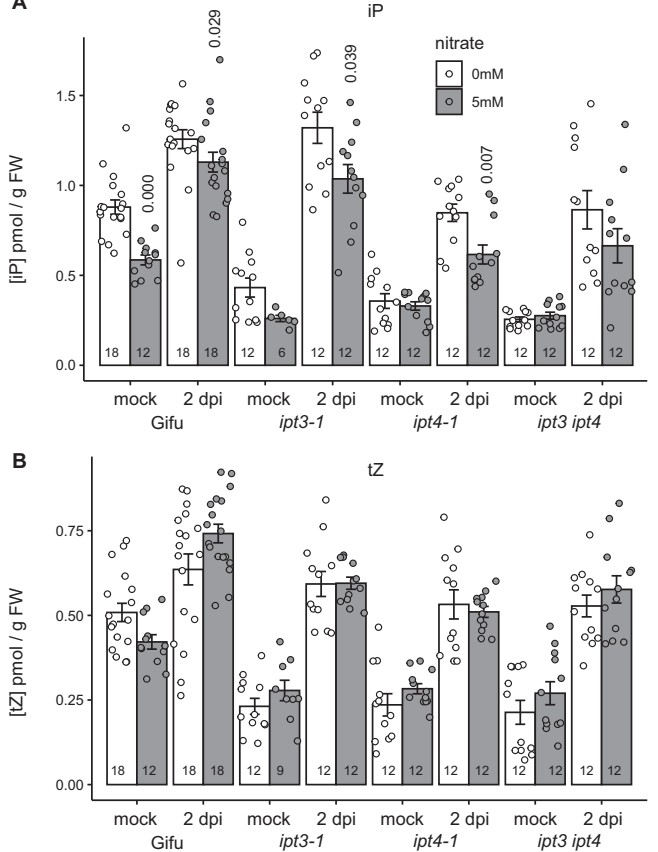

**Fig. 2 *ipt3* and *ipt4* mutants show reduced root iP content in high nitrate conditions. A, B** Cytokinin iP (**A**) and tZ (**B**) free base content analysed in mock or 2 days post inoculation with *M. loti* in the absence and presence of 5 mM KNO₃ using the same root tissue as illustrated in Fig. 1A. Bars show mean ± SE for the *n* value shown on each bar. Significant differences between nitrate presence and absence conditions as determined by Wilcoxon rank-sum test are indicated by the annotated *P* values.

To study whether *Ipt3* and *Ipt4* also play a role in resistance to nitrate inhibition of nodule initiation and rhizobia infection (IT), we counted the nodule primordia and infection threads (IT) that developed 7 dpi with *M. loti*. In the absence of nitrate, *ipt3-2 ipt4-1* nodule primordia formation is not significantly different from Gifu. However, in high nitrate conditions, *ipt3-2 ipt4-1* did not develop any nodule primordia at 7dpi, significantly less than the few visible primordia are already developed on Gifu (Fig. 4A).

In contrast to the positive role in nodule organogenesis, cytokinin plays a negative role in regulating infection by rhizobia[6]. Similar to the hyperinfection phenotype of the *lhk1* cytokinin receptor mutants[6], *ipt3-2 ipt4-1* forms significantly more IT than Gifu, implying cytokinin biosynthesis plays a negative role in IT formation (Fig. 4B). However, under high nitrate conditions, both Gifu and *ipt3-2 ipt4-1* show significantly reduced IT formation relative to nitrate-free plants (Fig. 4B).

**Cytokinin application rescues nitrate inhibition of nodule maturation and nitrogen fixation.** Because IPTs are key enzymes in cytokinin biosynthesis, we asked whether application of the cytokinin 6-Benzylaminopurine (BA) could rescue the hypersensitivity to nitrate inhibition seen in *ipt3-2 ipt4-1*. When grown in the presence of 10⁻⁸ M BA, Gifu develops shorter roots and forms less nodules compared with mock treatment (Fig. 5A–D and Supplementary Fig. 2). This inhibition of root

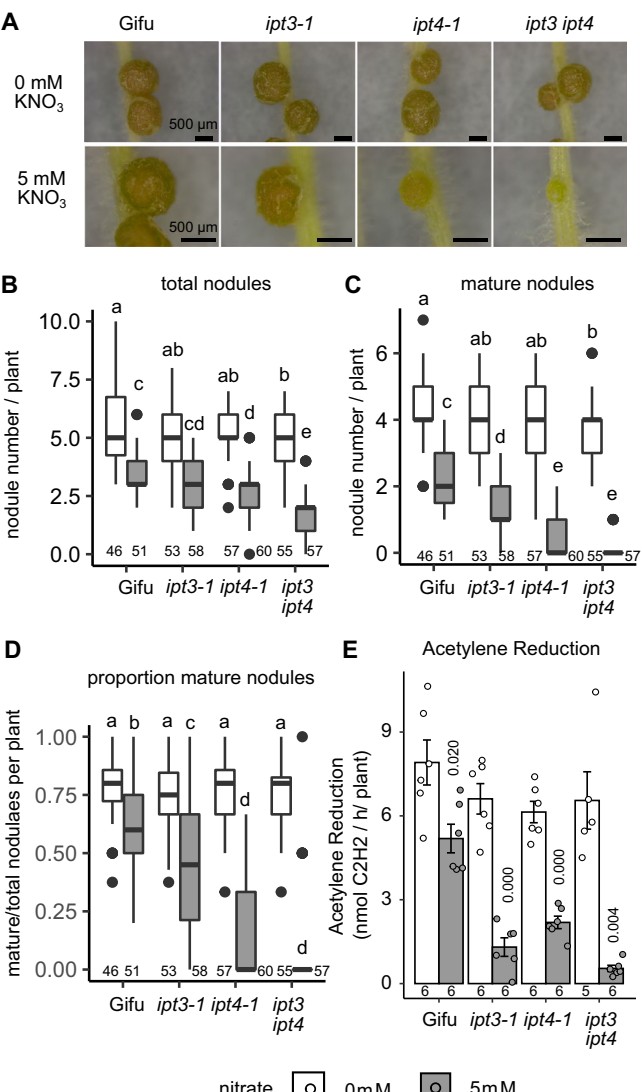

**Fig. 3 *Ipt3* and *Ipt4* are required for resistance to nitrate inhibition of nodulation. A** Images of nodules developed in the absence and presence of 5 mM KNO₃ at 14 dpi with *M. loti* on the indicated host genotypes. Scale bar = 500 μM. **B–D** Development of total (**B**), mature (**C**) and the proportion of mature nodules (**D**) in the absence and presence of 5 mM KNO₃ at 14 dpi with *M. loti*. **E** Nitrogenase activity assessed by Acetylene Reduction Assay (ARA) at 21 dpi with *M. loti* in the absence and presence of 5 mM KNO₃. Box plots show Min, Q1, Median, Q3, Max and outlier values in nodulation assays (**B–D**). Significant differences among different genotypes and nutrient conditions are indicated by letters (*p* < 0.05) as determined by ANOVA and Tukey post-hoc testing with pairwise *P*-values available in the Source Data file. For ARA (**E**), bars show mean ± SE and significant differences between Gifu and mutants are indicated by the annotated *P* values as determined by Student's *t* test. The *n* values of each group are shown.

elongation is also seen in *ipt3-2 ipt4-1* grown on 10⁻⁸ M BA (Supplementary Fig. 2). However, grown in the presence of BA (10⁻⁸ M BA), *ipt3-2 ipt4-1* increases to an average of 0.90 mature pink nodules under the high nitrate condition, which is significantly greater than the average 0.13 observed without added cytokinin (Fig. 5A–D). Applying BA at lower concentration (10⁻⁹ M BA) is also able to trigger formation of more total nodules on *ipt3-2 ipt4-1* in the presence of nitrate, while 10⁻⁸ M BA did not increase total nodule numbers (Fig. 5B). In wild type,

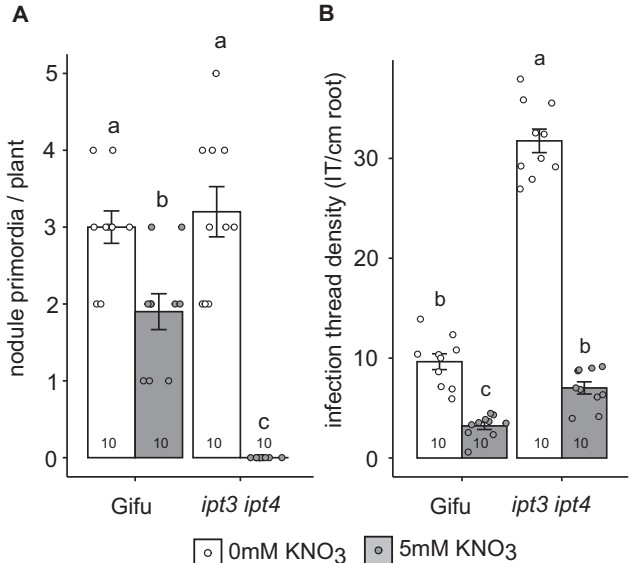

**Fig. 4 Ipt3 and Ipt4 contribute to maintenance of nodule initiation in the presence of nitrate. A** Nodule primordia number in the absence and presence of 5 mM $KNO_3$ at 7dpi with *M. loti*. **B** Infection thread density in the absence and presence of 5 mM $KNO_3$ at 7dpi with *M. loti*. Bars show mean ± SE with significant differences between genotypes and nutrient conditions indicated by letters ($p < 0.05$) as determined by ANOVA and Tukey post-hoc analysis. Pairwise *P*-values are available in the Source Data file. The *n* values of each group are shown.

further reductions in cytokinin application concentration ($5 \times 10^{-10}$ M BA), did not significantly impact nodule numbers (Supplementary Fig. 3).

We also measured nitrogenase activity by ARA in the BA rescued plants. Consistent with the pink nodule number, the ARA activity of BA-treated *ipt3-2 ipt4-1* is significantly higher than that in untreated plants in high nitrate conditions (Fig. 5E). Cytokinin application is thus able to rescue nitrate inhibition of nodule maturation and nitrogen fixation of the *ipt3-2 ipt4-1* mutant.

**Nitrate inhibition of cytokinin biosynthesis and signalling requires *Nlp1* and *Nlp4*.** NLP1 and NRSYM1 (here called NLP4) play central roles in nitrate signalling and nitrate inhibition of nodulation[22,28], thus we obtained LORE1 insertion lines[39] for each gene and characterised their phenotypes.

To assess the role of *Nlp1*- and *Nlp4*-mediated signalling in nitrate inhibition of symbiotic cytokinin biosynthesis, we analysed expression of cytokinin biosynthesis genes in the respective mutant backgrounds. As demonstrated in the earlier experiments, *Ipt2*, *Log1* and *Log4* are reduced by nitrate after *M. loti* inoculation (Fig. 6). Neither *nlp1-2* nor *nlp4-1* mutants show restriction of these cytokinin biosynthesis genes by nitrate, with *nlp4-1* having slightly higher *Log4* expression in high nitrate (Fig. 6A–C). In agreement with the biosynthesis gene expression, *nlp1-2* or *nlp4-1* mutants did not show nitrate-dependent changes in iP or tZ levels (Fig. 6F and Supplementary Fig. 4).

To confirm that these changes in cytokinin biosynthesis gene expression imparted meaningful changes on cytokinin signalling output, we also assessed the expression of two A-type response regulators in the same conditions. Both RR5 (LotjaGi4g1v0236600) and RR9 (LotjaGi1g1v0666300) showed a consistent inoculation response that is significantly dampened in nitrate-exposed roots of the wild-type plants (Fig. 6D–E). In contrast to wild type and consistent with the expression of cytokinin biosynthesis genes, neither of the response regulators

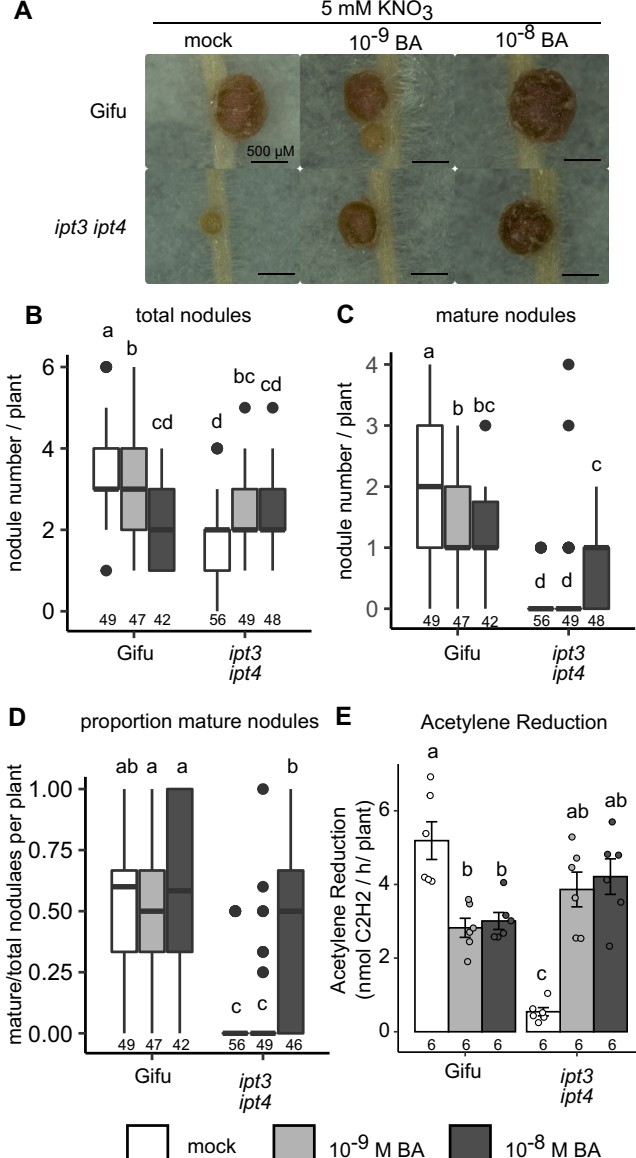

**Fig. 5 Nitrate inhibition of nodule development can be rescued by cytokinin application. A** Images of nodules developed in the presence of 5 mM $KNO_3$ with mock, $10^{-9}$ or $10^{-8}$ M BA at 14 dpi with *M. loti* on the indicated host genotypes. Scale bar = 500 μM. **B–D** Development of total (**B**), mature (**C**) and the proportion of mature nodules (**D**) in the presence of 5 mM $KNO_3$ with mock, $10^{-9}$ or $10^{-8}$ M BA at 14 dpi with *M. loti*. **E** Nitrogenase activity assessed by ARA at 21 dpi with *M. loti* in the presence of 5 mM $KNO_3$ with mock, $10^{-9}$ or $10^{-8}$ M BA. Box plots show Min, Q1, Median, Q3, Max and outlier values in nodulation assays (**B–D**). For ARA (**E**), bars show mean ± SE. Significant differences among different genotypes and concentration of BA are indicated by letters ($p < 0.05$) as determined by ANOVA and Tukey post-hoc testing. Pairwise *P*-values are available in the Source Data file. The *n* values of each group are shown.

were altered by nitrate treatment in the inoculated roots of *nlp1-2* nor *nlp4-1* plants.

In the absence of nitrate, *nlp1-2* forms slightly fewer nodules, while *nlp4-1* was not significantly different to wild-type Gifu (Fig. 7A–D). In high nitrate, where Gifu forms few pink nodules, *nlp1-2* and *nlp4-1* are able to form significantly more mature pink nodules, which is consistent with previous observations for *nlp* mutants in *M. truncatula* and *L. japonicus*[22,28]. Given the insensitivity to nitrate inhibition of

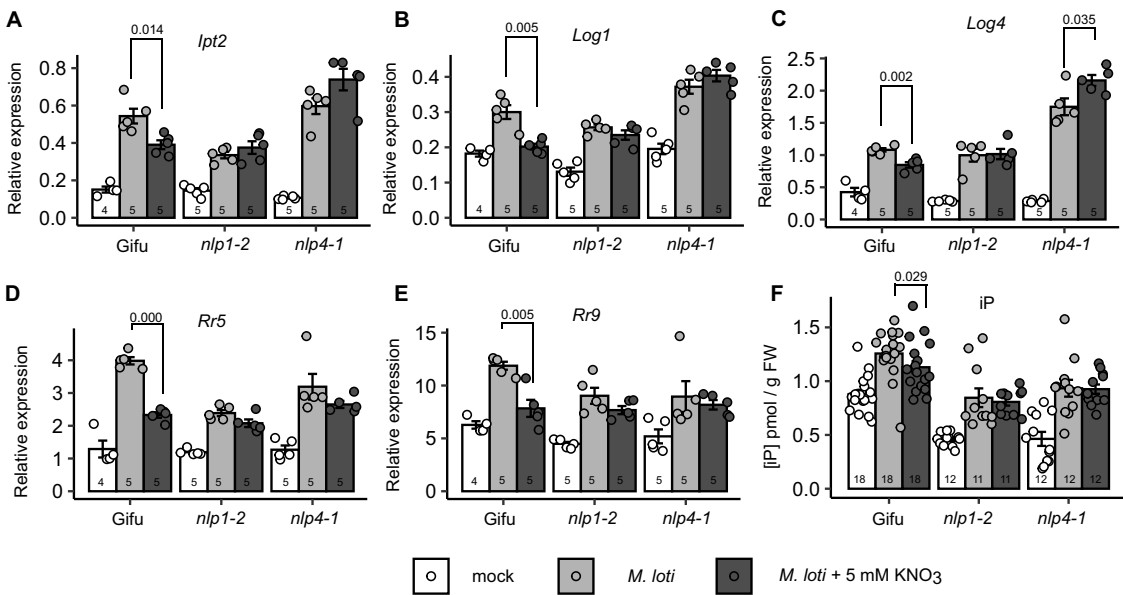

**Fig. 6 Nitrate restriction of cytokinin biosynthesis and signalling requires *Nlp1* and *Nlp4*.** Relative transcript abundance of *Ipt2* (**A**), *Log1* (**B**), *Log4* (**C**), *RR5* (**D**) and *RR9* (**E**) by qRT-PCR 1 dpi with *M. loti* in the absence and presence of 5 mM KNO$_3$ in the indicated genotypes. Ubiquitin is used as a reference gene. **F** Cytokinin iP free base content in the indicted genotypes at 2 dpi with *M. loti*. Bars show mean ± SE for the indicated *n* values. Significant differences between nitrate presence and absence conditions is indicated with annotated *P* values as determined by Student's *t* test in **A**–**E** and Wilcoxon rank-sum test in **F**.

cytokinin biosynthesis gene expression, we assessed the ability of *nlp1-2* and *nlp4-1* to rescue the nitrate sensitivity of the *ipt4* mutant. In high nitrate, *ipt4-1* shows significantly impaired nodule maturation with very few mature pink nodules compared with Gifu (Fig. 7A–D). However, *ipt4-1 nlp1-2* and *ipt4-1 nlp4-1* double mutants show nitrate-resistant nodulation in line with the *nlp1-2* and *nlp4-1* phenotypes, including developing mature nodules (Fig. 7A) in increased numbers when compared with *ipt4-1* or Gifu (Fig. 7B–D).

To assess the rescue of nodule function by *nlp* mutations, we also measured nitrogenase activity of these mutants by ARA at 21 dpi. In line with the nodule scoring, in the absence of nitrate, there was no significant difference between Gifu and any of the mutants (Fig. 7E). In high nitrate conditions, which inhibits nodule ARA activity compared with nitrate-free condition, *ipt4-1* exhibits a further reduction in ARA activity relative to Gifu (Fig. 7E). However, *nlp1-2* and *ipt4-1 nlp1-2* maintain equivalent ARA activity to nitrate-free conditions. Although *nlp4-1* and *ipt4-1 nlp4-1* exhibit lower ARA activity compared with nitrate-free condition, the nitrate inhibition of ARA activity are significantly less than in Gifu (Fig. 7E). To determine the nitrate sensitivity of *nlp* mutants in their cytokinin response, we conducted a spontaneous nodule assay by growing plants in the presence of 10$^{-8}$ M BA (Fig. 7F). Nitrate significantly reduces spontaneous nodule formation, which is consistent with previous reports[15], while *nlp1-2* mutants are resistant to this inhibition and *nlp4-1* retains nitrate sensitivity (Fig. 7F).

***Nlp1*- and *Nlp4*-mediated nitrate signalling inhibits cytokinin biosynthesis via interfering with NF signalling.** To identify nitrate-regulated genes that may contribute to the inhibition of symbiotic cytokinin biosynthesis, we exposed nitrogen-starved plants to high levels of nitrate (10 mM KNO$_3$) over a time series (0.25, 0.5, 1, 24 and 72 h) and conducted RNA-seq. Note that we used a higher nitrate concentration (10 mM) in the RNAseq to ensure a robust inhibition of nodulation signalling. Similar to the

rapid onset of primary nitrate responses in other species[40], we identified 425 genes that respond to nitrate within 15 min, and up to 4411 genes differentially regulated 72 h after nitrate exposure (Supplementary Fig. 5A). Among these genes, many nitrate marker genes, such as *Nrt2.1a*, *Nrt.1.b*, *Nia* and *Nir*, are induced at all time points in the series (Supplementary Fig. 5B). On the other hand, nitrogen starvation marker genes such as *Cep1* and *Cep7*[41] are suppressed after 24 h nitrate exposure (Supplementary Fig. 5B). Taken together, the general nitrate response in *L. japonicus* is similar to other species. However, in contrast to the induction of cytokinin biosynthesis, particularly *AtIPT3* by nitrate in Arabidopsis, none of the *LjIpt* or *LjLog* genes was significantly induced by nitrate across our time series (Supplementary Fig. 6).

Given symbiotic cytokinin biosynthesis is regulated by NF signalling[8,9], we analysed the expression of key components in early nodulation signalling, including NF signalling and downstream transcription factor genes under nitrate exposure. After 24 h nitrate exposure, *Nfr1*, *Nfr5, Nsp2* and *Nin* were all reduced, although the inhibitions of *Nfr1* and *Nfr5* did not persist after 72 h exposure (Fig. 8A). To investigate whether *Nlp*-mediated nitrate signalling plays a role in the nitrate restriction of these components, we investigated *Nfr1*, *Nfr5, Nsp2 Ern1* and *Nin* expression following nitrate exposure and rhizobia inoculation in Gifu, *nlp1-2* and *nlp4-1*. Consistent with previous RNAseq studies[42,43], rhizobia inoculation upregulates *Nsp2*, *Ern1* and *Nin* expression, while *Nfr1* and *Nfr5* expression is reduced (Fig. 8B–F). In turn, expression of *Nfr1*, *Nfr5, Nsp2*, *Ern1* and *Nin* were all significantly reduced in high nitrate in Gifu (Fig. 8B–F). In contrast, *nlp1-2* mutants showed no significant restriction of these genes in the high nitrate condition. In *nlp4-1* mutants, *Nfr1* expression was reduced by nitrate, while either no reduction (*Nfr5*, *Ern1*, *Nin*) or even greater induction (*Nsp2*) was found for the other genes in high nitrate conditions (Fig. 8B–F). Together, this indicates *Nlp1* and *Nlp4* mediate nitrate signalling repression of the upstream NF perception and signalling components in addition to their repression of cytokinin biosynthesis expression.

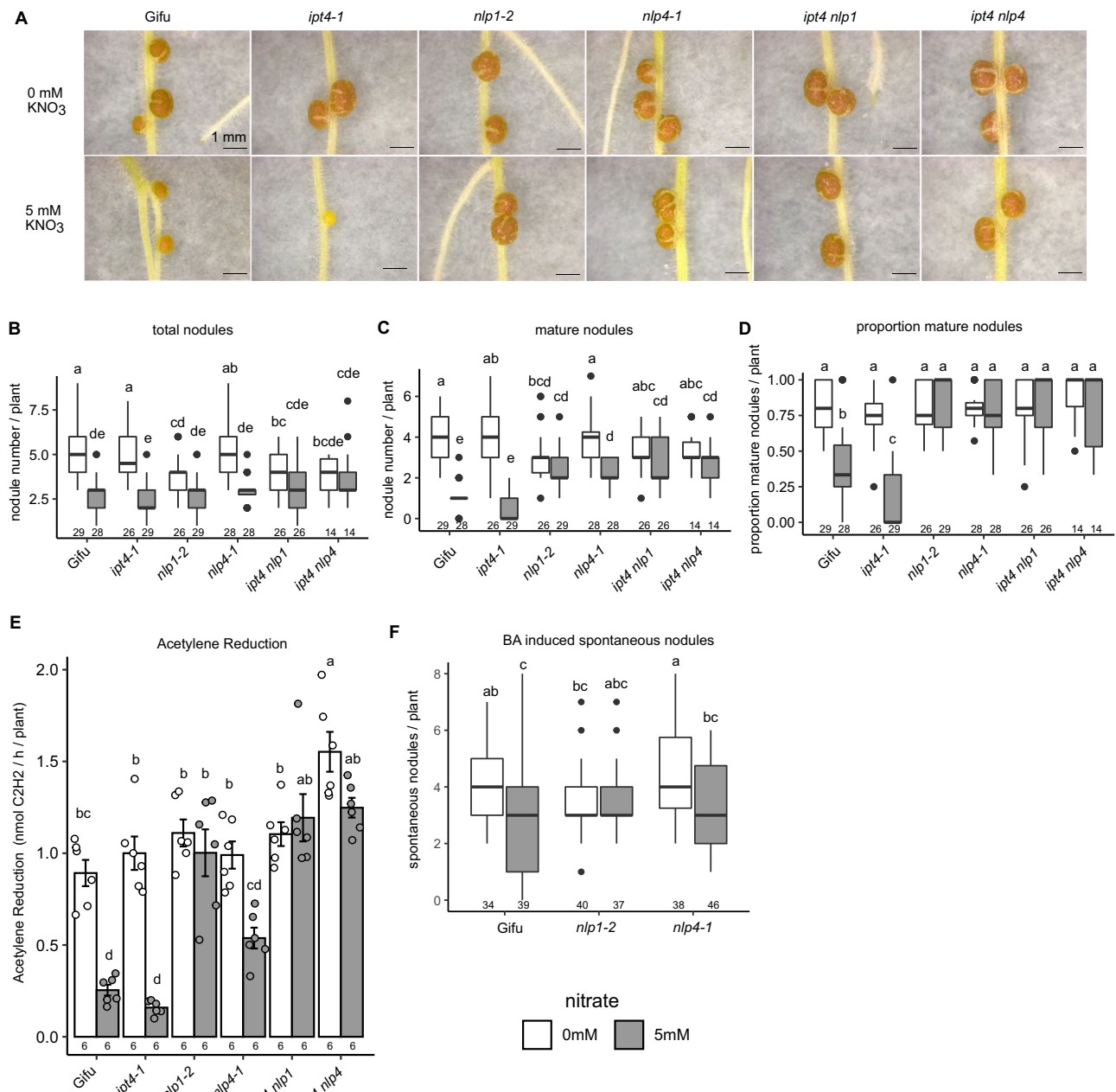

**Fig. 7 Nitrate signalling mediated by *Nlp1* and *Nlp4* acts upstream of cytokinin biosynthesis to restrict symbiotic signalling. A** Images of nodules developed in the absence and presence of 5 mM $KNO_3$ at 14 dpi with *M. loti* on the indicated host genotypes. Scale bar = 1 mm. **B–D** Development of total (**B**), mature (**C**), and proportion of mature nodules (**D**) in the absence and presence of 5 mM $KNO_3$ at 14 dpi with *M. loti*. **E** Nitrogenase activity assessed by ARA at 21 dpi with *M. loti* in the absence and presence of 5 mM $KNO_3$. **F** Formation of spontaneous nodules in response to cytokinin (BA). Box plots show Min, Q1, Median, Q3, Max and outlier values in nodulation assays (**B–D**, **F**). Bars show mean ± SE. Significant differences among different genotypes and nutrient conditions are indicated by letters ($p < 0.05$) as determined by ANOVA and Tukey post-hoc testing with pairwise *P*-values available in the Source Data file. The n values are shown below the respective groups.

## Discussion

In this study, we show that the symbiotic induction of cytokinin biosynthesis in *L. japonicus* is reduced by high nitrate concentrations, thereby reducing the positive influence of cytokinin on nodule organogenesis. Cytokinin biosynthesis mutants, which have reduced cytokinin levels exacerbate this response and are more sensitive to nitrate inhibition of nodule development. In agreement with this, we show that this increased nitrate sensitivity can be rescued by supplementing growth media with cytokinin. We also find that *Nlp1*- and *Nlp4*-mediated nitrate signalling is required for this reduction of symbiotic cytokinin

biosynthesis and nodule organogenesis, by suppressing the expression of upstream signalling components including *Nfr1*, *Nfr5*, *Nsp2*, *Ern1* and *Nin*.

In response to NF perception, many cytokinin biosynthesis genes are upregulated in the root susceptible zone where cytokinin accumulates transiently to trigger nodule initiation[8,9]. The cytokinin trigger for nodulation requires signalling via the receptors (predominantly *Lhk1/Cre1*), with loss-of-function mutants impaired in nodule development[5,6,11,12]. In contrast, cytokinin application or constitutive activation of cytokinin signalling can activate nodule organogenesis programs in the

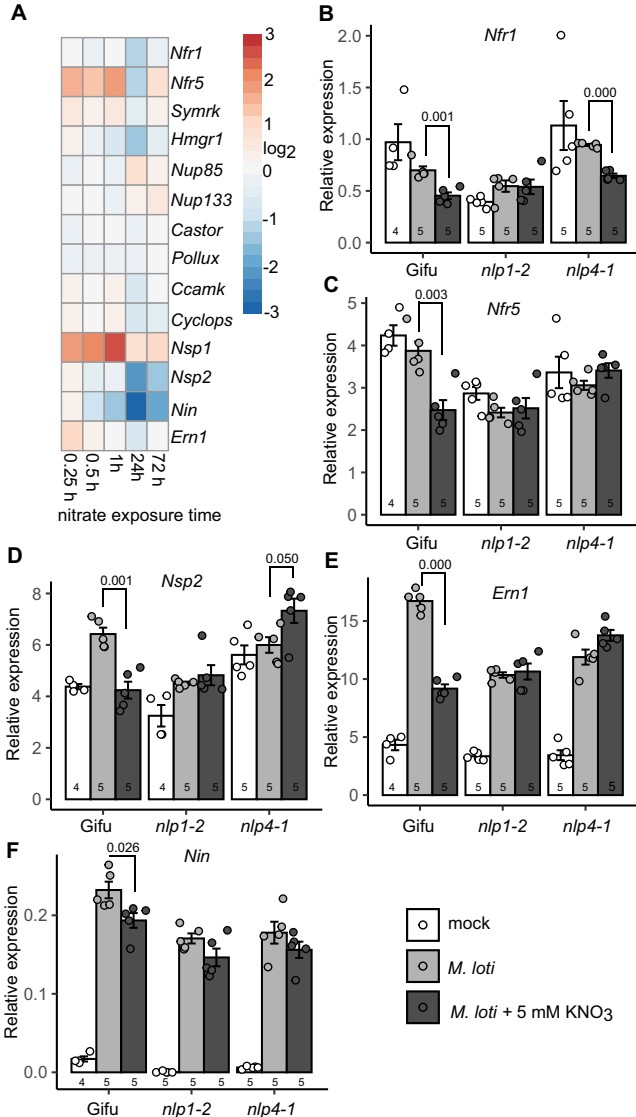

**Fig. 8 Nitrate inhibits symbiotic signalling via *Nlp1* and *Nlp4*. A** A heatmap of NF signalling gene expression at different times following nitrate exposure. **B–E** Relative transcript abundance of *Nfr1* (**B**), *Nfr5* (**C**), *Nsp2* (**D**), *Ern1* (**E**) and *Nin* (**F**) by qRT-PCR 1 dpi with *M. loti* in the absence and presence of 5 mM KNO₃ in the indicated plant genotypes. Ubiquitin is used as a reference gene. Bars show mean ± SE for for the indicated *n* values. Significant differences between nitrate presence and absence conditions is annotated with *P* values as determined by Student's *t* test. Gene IDs and expression values are shown in Supplementary Table 4.

absence of rhizobia[7,15,16]. Here, we find the transient induction of cytokinin in the susceptible zone is reduced when grown in high nitrate conditions. Nodule initiation and maturation, which is required for nitrogen fixation, is therefore suppressed and delayed by nitrate exposure. The inhibition of cytokinin biosynthesis genes we detected occurs in genes that are upregulated during symbiosis. This indicates that the reduction is likely, at least in part, to reflect the impact of nitrate on the nodulation signalling pathway. However, the reduced cytokinin levels detected in both inoculated and uninoculated plants supports the view that nitrate restriction of cytokinin biosynthesis in *L. japonicus* is enacted through both nodule-dependent and nodule-independent signalling processes.

In addition to the mechanism we describe where inhibited cytokinin synthesis leads to reduced nodule development,

excessive cytokinin accumulation can also inhibit nodulation. This has been shown to occur through induction of Auto-regulation of Nodulation (AON) and ethylene signalling, which both play negative roles in nodulation[44–48]. This is also supported by the enhanced sensitivity to nitrate inhibition of nodulation that occurs in the *Ljckx3* cytokinin over-accumulation mutants[49]. We find support for this fine balance of cytokinin for nodule development with lower concentrations of cytokinin supplementation able to rescue the biosynthesis mutant impairment, or at increased levels to inhibit nodulation. Together these observations support a model where cytokinin levels are tightly regulated by multiple internal and external influences to balance the requirements for plant nitrogen between uptake from the soil and nodule development (and thus nitrogen fixation).

Multiple mechanisms act on this pool to finely tune cytokinin signalling, including regulation of cytokinin biosynthesis. While cytokinin biosynthesis appears to be a redundant process in optimal nodulation conditions (lack, or only minor phenotypes, for *Ljipt3* and *Ljipt4*), under nitrate conditions that restrict nodulation, this redundancy is only partial. Previous studies of *ipt3* have reported variable phenotypes (reduced nodulation[13], no phenotype[8], or increased nodulation[14]), and may support the role of *LjIpt3* as an environment-dependent regulator of nodule development. The requirement for finely tuned cytokinin output is also exemplified by reduced nodule development in nitrate-exposed *Ljipt3* mutants, despite no evident reduction in iP levels. The phenotype in this case may result from a reduced overall cytokinin level in this mutant in combination with nitrate attenuation of additional downstream cytokinin signalling components.

Reduced cytokinin biosynthesis in high nitrate conditions impacts the number of mature nodules, as well as nitrogen fixation activity. Whether this reduction in nitrogen fixation is a direct effect of cytokinin, or is indirectly related to the failure of nodules to mature remains to be determined. Supporting an indirect effect is the apparent reduction in size of nodules in high nitrate conditions with the onset of fixation only occurring when nodule growth and maturation is sufficiently stimulated by cytokinin. Feedback "sanctioning" in response to reduced nitrogen fixation activity is also known to restrict nodule size[50]. A more direct regulation of nitrogen fixation by cytokinin could then trigger such regulatory mechanisms, restricting nodule growth.

In contrast to the requirement in nodule organogenesis, cytokinin plays a negative role in rhizobia infection[6,51]. Consistent with the hyperinfection phenotype of *Ljlhk1*, the *ipt3-2 ipt4-1* double mutant shows more than triple the infection density of wild type. This shows that cytokinin biosynthesis is required for negative regulation of rhizobial infection. Cytokinin shows extensive crosstalk with ethylene signalling, including through stabilisation of ACC synthase to promote biosynthesis[52]. The hyperinfection phenotypes of *Ljlhk1* (and presumably *ipt3 ipt4* mutants) are thought to stem, at least in part, from reduced stimulation of ethylene signalling[51]. In high nitrate conditions this hyperinfection is reduced in *ipt3-2 ipt4-1*, with a similar degree of inhibition to wild type. This implies that at least cytokinin biosynthesis is not critical to nitrate inhibition of rhizobia infection. The ethylene and AON pathways may therefore play more prominent roles in negatively regulating infection by nitrate, independent of cytokinin. In addition, *MtNlp1* may directly interfere with infection by blocking NIN's function[28]. Thus, nitrate inhibition of rhizobia infection is likely to target alternative pathways or downstream of cytokinin biosynthesis.

In non-legumes, cytokinin biosynthesis plays an important role in signalling root nitrate availability and coordination of shoot growth[35,53]. This cytokinin response to nitrate has been reported

in several species including *Arabidopsis*[37], rice[32] and maize[31]. In response to nitrate, *t*Z and iP type cytokinin accumulate in the shoot, but cytokinin level remains unchanged in roots of *Arabidopsis*[35]. Here we show that in *Lotus japonicus* root susceptible zone, where cytokinin plays an important role in symbiotic organ establishment, cytokinin biosynthesis is not enhanced, but rather reduced by nitrate supply. This reduction was evident in both reduced expression of biosynthesis genes (after inoculation) and reduced cytokinin levels in both uninoculated and inoculated roots. While the reduction in cytokinin, in particular iP, is consistent, the mechanism underlying the response in uninoculated and inoculated roots shows some differences. In inoculated roots, where cytokinin synthesis is strongly stimulated, restriction of the symbiotic pathway impacts expression of multiple cytokinin biosynthesis genes. In uninoculated roots, we did not find significant inhibition of the same genes, despite the significant reduction in cytokinin levels. This suggests other mechanisms, such as increased cytokinin degradation or translocation, may occur in uninoculated roots exposed to high nitrate.

*L. japonicus* may therefore signal nitrate availability using alternative pathways to cytokinin signalling, which is prominent in non-legumes. One possibility is that *L. japonicus* maintains a cytokinin response to nitrate in non-susceptible root tissue, with the nitrate-reduced and rhizobia-enhanced cytokinin being restricted to root zones supporting nodulation. Supportive of this is that nitrate can only induce AtIpt3 in vascular bundles[37] which differs from the predominantly cortical location of symbiotic cytokinin in *Lotus*[8]. In soybean for example *GmIpt3* and *GmIpt15* are induced by nitrate[54], and the location of this induction and influence on nodule development would be interesting to determine. Alternatively, specific roles for iP and *t*Z cytokinin types may play a role, as we only identify significant reductions in iP by nitrate, which has a net result of increasing the ratio of *t*Z to iP. Such a model would imply that iP is predominantly required for nodule development in the root, while *t*Z participates in signalling nitrate availability. Both cytokinin-dependent and -independent pathways are thought to play roles in coordinating systemic nitrogen signals[55]. Other root–shoot coordination pathways such as CEP signalling[56,57] may therefore play a more prominent role in coordinating shoot growth with nitrogen availability in legumes, relegating cytokinin signalling of nitrate to a more minor role.

Nitrate negatively influences many processes in nodulation, including NF signalling, nodule initiation and development, rhizobia infection and nitrogen fixation[58]. *Nlp*s play central roles in nitrate signalling and development in many species[26,59]. Both *LjNlp1*, *LjNlp4*, and orthologues in *Medicago* are involved in nitrate inhibition of nodule initiation and development, rhizobia infection and nitrogen fixation[22,28]. *LjNlp4*, in response to nitrate, directly targets *LjCle-rs2* to trigger the AON pathway and restriction of nodule initiation[22]. *Mt*Nlp1 and *Lj*Nlp4 are able to interact and/or compete with Nin to block Nin function, which is essential for nodule initiation and development and rhizobia infection[28,29]. We found that nitrate restriction of NF receptors expression, which is likely to reduce the susceptibility to rhizobia, requires *Nlp*s. Additionally, we found *Nlp*s are also required for nitrate restriction of some nodulation transcription factors, such as *Nsp2*, *Ern1* and *Nin* (Fig. 9). In addition to this nitrate-dependent regulation, we observed a general reduction in symbiotic gene expression in *nlp* mutants, consistent with a recent demonstration of their role in symbiotic gene regulation through cooperative regulation at *Nin* binding sites[29]. Taken together with previous studies, *Nlp*s regulation of nodule initiation and development, rhizobia infection and nitrogen fixation is likely to occur

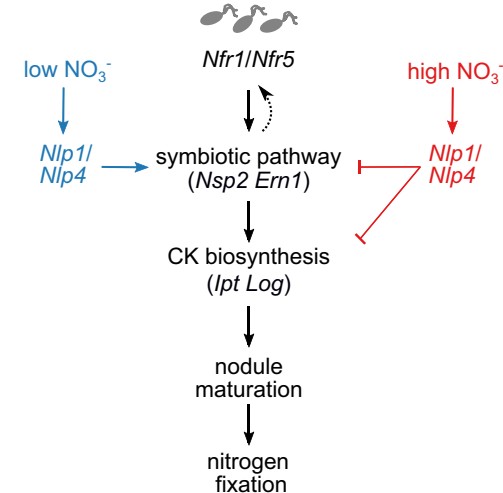

**Fig. 9 Proposed model for regulation of nodule organogenesis by nitrate.** In low nitrate, NLP signalling contributes to regulation of symbiotic genes, which stimulate cytokinin biosynthesis, ensuring robust nodule maturation and onset of nitrogen fixation. In high nitrate, NLP-dependent nitrate signalling inhibits expression of symbiotic genes and symbiotic cytokinin biosynthesis that is crucial for nodule organogenesis. Nodulation is thus balanced between a permissive low-nitrate state where high cytokinin levels can accumulate, or a restrictive state with low root cytokinin.

through restriction of key components in NF perception and signalling. Alternatively, cytokinin may function to regulate *Nlp1* and *Nlp4* action. Cytokinin induction of the transcription factor *Nin*, which antagonises NLP function, may provide such a link[60]. This restriction of symbiotic gene expression by the NLP-dependent pathways, alongside the reduction in symbiotically triggered cytokinin biosynthesis, corresponds to a reduction in nodule development (Fig. 9). Our finding that cytokinin application can rescue some inhibitory effects of nitrate on nodule maturation and nitrogen fixation, together with the nitrate sensitivity of spontaneously formed nodules, shows that both biosynthesis and downstream cytokinin signalling are regulated by nitrate during nodule development. Further mechanistic studies into how *Nlp*s mediate nitrate regulation and interact with other nodule regulatory pathways to finely tune nodulation in response to the environment and plant resource requirements may allow manipulation to improve nitrogen fixation and yield in an economic and sustainable way.

In conclusion, we propose a model where nitrate interferes with NF signalling and symbiotic cytokinin biosynthesis in the root susceptible zone, ultimately suppressing nodule organogenesis (Fig. 9). In high nitrate, *Nlp1* and *Nlp4* restrict the early symbiotic pathway components, resulting in less sensitivity to NF and reduced output of NF signalling. This reduced NF signalling capability inhibits cytokinin accumulation, which is essential for nodule initiation and development, essential prerequisites for nitrogen fixation in mature nodules.

## Methods

**Plant and bacteria genotypes**. *Lotus japonicus* ecotype Gifu was used as wild type[61], while *ipt3-1*, *ipt4-1* and *ipt3-2 ipt4-1* were LORE1 insertion mutants[8]. LORE1 mutants *nlp1-2* and *nlp4-1* were ordered through LotusBase (https://lotus.au.dk) and homozygotes were isolated for mutants as described[39]. Plants carrying LORE1 insertions were identified from segregating populations using a primer specific to the LORE1 insert (Table S2). Homozygotes were identified using primers flanking the insert position. Line numbers and genotyping primers are given in Table S2. *Mesorhizobium loti* NZP2235 was used for nodulation assay. For infection thread observation, *M. loti* R7A strain constitutively expressing DsRed was used.

**Plant and bacteria growth conditions**. Phenotyping on growth plates was conducted by transferring 3-day-old seedlings onto filter paper placed on vertical 1.4% agar noble plates containing ¼ Long Ashton (Table S3) in the presence of 5 mM KCl or KNO₃. For the cytokinin rescue assay, 6-Benzylaminopurine (BA, Sigma-Aldrich) were added into ¼ Long Ashton plates containing 5 mM KNO₃. Three days after transfer, seedlings were inoculated with rhizobia inoculum OD$_{600}$ = 0.015. Infection threads were counted at 7 day post inoculation (dpi) using a Zeiss axioplan fluorescence microscope, while nodule numbers were counted at 14 dpi. For nitrate and RNA-seq, 3-day-old seedlings were transferred on ¼ B&D plates and grown for 14 days.

**Gene expression analysis**. For RNA-seq, following 14 days growth on nitrate-free plates, plants were acclimatised prior to treatment by submerging in ¼ B&D liquid medium overnight, then treated with 10 mM KNO₃ for 0, 0.25, 0.5, 1, 24 or 72 h. Root tips were removed (3 mm from tip) and the remainder of the root harvested. mRNA were isolated using the NucleoSpin RNA Plant kit (Macherey-Nagel) then library preparation and PE-150 bp Illumina sequencing was conducted by Novogene.

For qPCR, the root-susceptible zone was harvested at 1 or 2 dpi. mRNA were isolated using the kit described above. RevertAid Reverse Transcriptase (Thermo) was used for the first strand of cDNA synthesis. LightCycler480 SYBR Green I master (Roche Diagnostics) and LightCycler480 instrument were used for the real-time quantitative PCR. Ubiquitin-conjugating enzyme was used as a reference gene[62]. The initial cDNA concentration of each target gene was calculated using amplicon PCR efficiency calculations using LinRegPCR[63]. Target genes were compared with the reference genes for each of 5 biological repetitions (each consisting of 6 to 10 plants/root susceptible zones). At least two technical repetitions were performed in each analysis. Primers used are listed in Supplementary Table 1.

**Cytokinin extraction, detection and quantification**. For cytokinin extraction from *L. japonicus* material, ~20 mg of snap-frozen root and nodule material was used per sample and analysed[Citation error] as previously described[64]. The tissue was ground to a fine powder using a TissueLyser and metal beads (Qiagen, Germantown, MD, USA). Ground samples were extracted in 1 mL 100% methanol (MeOH) containing stable isotope-labelled internal standards (IS) at 100 nM per compound. After vortexing and ultrasonication for 30 s, extraction was performed with shaking at 4°C overnight. Samples were centrifuged at 12,000 rpm for 10 min at 4°C. Supernatants were transferred to 4-mL amber glass vials. 1 mL of 100% MeOH was used to re-extract pellets for 1 h at 4°C. After centrifugation, both extractions were pooled before evaporation to dryness in a speed vac (SPD121P, ThermoSavant, Hastings, UK). Formic acid (1 mL, 1 M) was used to elute samples before loading onto a 30 mg Oasis MCX Cartridge (Waters, Milford, OH, USA). Each cartridge was washed with 1 mL of MeOH and equilibrated with 1 mL of 1 M formic acid prior to loading. The loaded cartridge was washed with 1 mL of 0.35 M NH₄OH, and then eluted with 1 mL of 0.35 MnH₄OH (in 60% MeOH). The 0.35 M NH₄OH was evaporated by speed vac at RT, and the residue stored at −20°C until further analysis.

For quantification, samples were resuspended in 100 µL of methanol/water (0.1% formic acid) (1:9, v/v) and filtered through a 0.45 mm Minisart SRP4 filter (Sartorius, Goettingen, Germany). Analysis of cytokinins was performed by comparing retention times and mass transitions to the unlabelled standards, using a Waters Xevo TQs mass spectrometer equipped with an electrospray ionisation source coupled with Acquity UPLC system (Waters, Milford, OH, USA). Chromatographic separations were conducted using an Acquity UPLC BEH C18 column (100 mm, 2.1 mm, 1.7 mm; Waters, Milford, OH, USA) by applying a methanol/water (0.1% formic acid) gradient. The column was operated at 40°C with a flow rate of 0.25 mL min$^{-1}$, and was equilibrated for 30 min using a methanol/water (0.1% formic acid) (5:95, v/v) composition at the start of the run.

The methanol/water (0.1% formic acid) gradient started from 5% (v/v) methanol, increasing to 70% (v/v) methanol in 17 min. To wash the column, the water/methanol gradient was increased to 100% methanol in a 1 min gradient, then maintained for 1 min before returning to 5% methanol using a 1 min gradient, prior to the next run. Volume of sample injection was 5 µL. Cone and gas flows were 150 and 800 L h$^{-1}$, respectively. The capillary voltage was set at 3.0 kV, source temperature 150°C and desolvation temperature 550°C. The cone voltage was optimised for each standard compound using the IntelliStart MS Console (Waters, Milford, OH, USA). Argon was used for fragmentation. Multiple reaction monitoring (MRM) was used for quantification. Parent–daughter transitions for the different (stable isotope labelled) compounds were set using the IntelliStart MS Console. The cone voltage was set to 40 eV. To determine sample concentrations, a 10-point calibration curve was constructed for each compound ranging from 0.1 µM to 19 pM, in addition to a known amount of deuterium-labelled internal standard.

**Acetylene reduction assay**. Acetylene reduction assays were conducted essentially as described previously[49]. The nodulated root from single plants was placed in a 5 ml glass GC vial. A syringe was used to replace 250 µl air in the vial with 2% acetylene. Samples were incubated for 30 min before ethylene quantification using a SensorSense (Nijmegen, NL) ETD-300 ethylene detector operating in sample mode with 2.5 L/h flow rate and 6-min detection time. The curve was integrated using the SensorSense valve controller software to calculate the total ethylene production per sample.

**Statistical analysis**. Statistical analyses were carried out using R software (R Core Team, 2015). All statistical testing was two-sided. Comparison of multiple groups included ANOVA followed by Tukey posthoc testing to determine statistical significance indicated by different letter annotations. All pairwise P-values are included in the Source Data file. When making single comparisons, Student's t test was used as indicated, while non-parametric Wilcoxon rank-sum testing was used for cytokinin measurements which were not normally distributed due to the presence of outliers in some groups.

For statistical analysis of RNAseq, a decoy-aware index was built for Gifu transcripts using default Salmon parameters and reads were quantified using the -validateMappings flag (Salmon version 0.14.1)[65]. Expression was normalised across all conditions using the R-package DESeq2 version 1.20[66] after summarising gene level abundance using the R-package tximport (version 1.8.0). Normalised count data for all genes is shown in Supplementary Data 1. Differentially expressed genes with correction for multiple testing were obtained from this DESeq2 normalised data and are shown in Supplementary Data 2.

**Reporting summary**. Further information on research design is available in the Nature Research Reporting Summary linked to this article.

## Data availability
RNAseq raw data has been submitted to NCBI under BioProject accession number PRJNA642098. Cytokinin measurements have been deposited at Metabolights under submission number MTBLS3492. Source data are provided with this paper with all raw data and results of statistical tests used for producing figures available in the Source Data file. Source data are provided with this paper.

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

## Acknowledgements

We thank Finn Pedersen for expert plant handling and Francel Verstappen for assistance with cytokinin quantification. We are grateful for support from the project Engineering Nitrogen Symbiosis for Africa (ENSA) currently supported through a grant to the University of Cambridge by the Bill & Melinda Gates Foundation and UK government's Department for International Development (DFID). Analysis by Y.R. and W.K. is supported by Netherlands Organization for Scientific Research (VENI863.15.010).

## Author contributions

J.L. conceived and performed experiments, analysed data, prepared figures and wrote the paper. Y.R. performed cytokinin quantification. W.K. designed cytokinin quantification procedure and analysed the data. J.S. conceived experiments, supervised and revised the paper. D.R. conceived experiments, analysed data, supervised and wrote the paper.

## Competing interests

The authors declare no competing interests.
