## [Peer Review File · Nature Communications]

REVIEWER COMMENTS

Reviewer #1 (Remarks to the Author):

The manuscript reports an important finding that nitrate inhibition of nodulation occurs through regulation of cytokinin biosynthesis. The authors show that expression of cytokinin biosynthesis genes and levels of cytokinin (iP and tZ) are reduced in the root susceptible zone under high nitrate conditions. They also show that cytokinin biosynthesis mutants are hypersensitive to nitrate inhibition of nodulation and that application of cytokinins can rescue this hypersensitivity. Finally they show that Nlp1 and Nlp4 are required for the regulation of cytokinin biosynthesis genes by nitrate and that a number of nod factor signaling components are regulated by nitrate via Nlp1 and Nlp4. The paper reports a comprehensive evaluation of how cytokinin biosynthesis is regulated by nitrate during nodulation. The paper is generally well-written. I have the following comments.

1. The authors should clearly distinguish conclusions based on direct regulation of cytokinin biosynthesis by nitrate versus regulation in the presence of rhizobia. This is important since nitrate also suppresses nod factor signaling which can indirectly lead to reduced cytokinin biosynthesis. For example, the expression of Ipt4 and Log4 and the levels of iP seem to be reduced by nitrate in mock inoculation samples (direct regulation) whereas Ipt2 and Log1 expression is reduced by nitrate only under rhizobium inoculation.

2. Conclusions on regulation of nodule maturation independent of nodule initiation could be improved by evaluation what proportion of nodules were mature instead of the number of mature nodules (Lines 134-136).

The figure legend in Figure 3, 5 and 7 are incorrect. Panel B shows the number of total nodules and panel C shows the number of mature nodules.

3. Did the authors calculate ARA on a per nodule basis? Is nitrogen fixation inhibited per se by nitrate or is the reduced ARA simply due to reduced number of nodules.

4. Lines 158-159: The statement is very confusing. Revise it for clarity e.g. " ipt3-2 ipt4- form significantly more nodule primordia than Gifu".

5. Line 163: Consider revising "more than 3 times more"

6. Line 167 and associated text: Consider revising "nitrate resistance to nodule initiation..."

7. Line 411: Provide specifics; How many millimeters of root tip was removed?

Reviewer #2 (Remarks to the Author):

The manuscript submitted by Lin et al reports the role of cytokinin, which is important for regulation of nodulation, in the nitrate inhibition of nodulation. The authors took advantage of *Lotus japonicus* LORE1 insertion mutants, and examined nodulation of mutants in response to exogenous nitrate and cytokinin. Notably, they found that suppression of nodulation, caused in cytokinin biosynthesis mutants under inhibitory nitrate condition, was overcome by mutation in NLP, which is a central regulator of nitrate responses. The nitrate inhibition of nodulation was further examined in the expression of symbiosis genes that are involved in NF signaling. The authors concluded that nitrate suppresses the NF signaling, resulted in reduction of cytokinin biosynthesis, thereby attenuates nodule initiation and development.

Overall, the manuscript is comprehensive to read, the experiments are precisely described, and publications are properly cited. I have however several concerns as written below.

1. From the double mutant analysis shown in Fig. 7, it is genetically clear that nlp1 (in the case of

mature nodules) and *nlp4* are epistatic to *ipt4*, thereby CK (or cytokinin biosynthesis at least) inhibits the NLP function in suppression of nodulation by nitrate. This fits very well into the results in Figs. 3 and 5, but is totally contradictory to the title and the conclusion. I understand from Fig. 6 that NLPs are involved in the nitrate inhibition of the expression of cytokinin biosynthesis and signaling genes. Therefore, from the results shown in the manuscript two pathways should be equally considered.

2. There can be several experiments that authors may consider in this context: One is application of CK to *nlp* mutants. If nitrate inhibits nodulation through inhibition of cytokinin biosynthesis, the effect of CK in (spontaneous) nodulation should be nitrate insensitive, contrary to the situation in WT as shown in Heckmann et al. 2011. In addition, since the gain-of-function LHK1 mutant *snf2* is nitrate sensitive (Tirichine et al. 2006), combination of *snf2* and *nlp* mutants should be nitrate insensitive as well.

3. The conclusion drawn from Fig. 8 is overstatement (description started from L. 290, L. 369, L. 379, Fig. 9, and the section title in L. 248). It is true from the results that reduction in the expression of symbiosis genes by nitrate is NLP dependent, but this degree of reduction in WT is not validated for suppression of nodulation, or for inhibition of cytokinin biosynthesis, by nitrate. Please rephrase texts appropriately. In addition, according to Fig. 8A, *Nin* is the most notable gene which expression is downregulated by nitrate. Please include *Nin* expression in WT and *nlp* mutants alongside with other symbiosis genes.

Specific comments:

1. L. 30 "... nodule development, maturation and nitrogen fixation.", L. 31 "... nodulation and nitrogen fixation ...": To be precise, authors did not show (the activity of) nitrogen fixation per se, because plants were grown in the presence of nitrate for 3 weeks (Figs. 3 and 5). As authors mentioned in the text (L. 139), this is proxy of nodule maturation, not nitrogen fixation. Please rephrase the sentences.

2. L. 46 "... cytokinin synthesis and signalling.": Cytokinin transport is equally important for nodulation. Please refer Jarzyniak et al. 2021 (doi: 10.1038/s41477-021-00873-6).

3. L. 54 "The AON pathway thus integrates ... soil nitrate availability.": There is no description on the AON pathway related to nitrate so far. Please describe some in the paragraph.

4. L. 61 and thereafter as necessary: Please also refer Nishida et al. 2021 (doi: 10.1093/plcell/koab103) for NLPs in nodulation.

5. Figure 1: Why did authors omit CYP735A? They previously showed that the expression of CYP735A was induced after inoculation (Reid et al. 2017), and moreover, there is a marked difference in abundance between *iP* and *tZ* upon nitrate treatment. Since CYP735A is responsible for conversion of precursors of *iP* to those of *tZ*, it is expected that nitrate influences CYP735A expression. Please add the data.

6. Figure 2 and thereafter: Why did authors omit *ipt2* mutants for analysis? They mentioned that IPT2 is "the major player in elevated cytokinin responses from the IPT family" (Reid et al. 2017). Absence of LORE1 insertion mutants (as mentioned in Reid et al. 2017) is not a rational excuse, because nowadays knockout by CRISPR is feasible even in *Lotus japonicus*. Please involve *ipt2* mutants in the analysis.

7. Figure 2: The activity of IPT affects both *iP* and *tZ* contents. Please measure *tZ* as well.

8. Figure 5: Application of exogenous CK reduces total and mature nodules in WT. This is fine, as authors mentioned that "excessive cytokinin accumulation can also inhibit nodulation" (L. 304). However, on the assumption that "nitrate inhibits nodule organogenesis through inhibition of cytokinin biosynthesis" as shown in the title, lower concentration of exogenous CK should be tested to know whether it is also true in WT.

9. Figure 6: I understand that attenuation of the expression of cytokinin biosynthesis and signaling genes by nitrate requires NLP1 and NLP4, but at the same time I wonder whether it is also applicable to *iP* and *tZ* contents. This is important because some of the genes are more abundant in the mutant background, indicating higher CK accumulation. Please measure *iP* and *tZ* as shown in Fig. 1.

10. Figure S3c: Some of the datapoints are not the natural numbers (i.e. 2.5 or 7.5). Are values average? If not, please revisit the data.

Reviewer #3 (Remarks to the Author):

This is an important work that provides a mechanistic explanation for a long recognized but not well understood phenomenon of nitrate inhibition of root nodule formation in legume plants. Legumes, unlike most other plants, can tap into atmospheric nitrogen to support their growth, independent of soil nitrogen. They do so by forming root nodules that host symbiotic, nitrogen fixing bacteria. However, soil nitrogen is inhibitory and high concentrations of KNO₃ restrict nodule development. Understanding the underlying process has been a long standing goal in the effort to enhance nitrogen fixation in legumes and also to improve nitrogen uptake/use efficiency, an important agronomic trait, in non-legume crop species.

The authors now show that the main target of the nitrate-dependent regulation is the symbiosis pathway, which is attenuated by a NIN-like protein-dependent mechanism, leading to a local cytokinin deficiency that restricts nodulation. The corollary hypothesis, based on their observations, is that under nitrogen limited conditions, legumes differ from non-legumes in their cytokinin biosynthesis response to nitrate. Their results constitute a significant contribution which should be of considerable interest to a broad audience, including evolutionary and developmental biologists. The data will also be of specific interest to researchers working in the symbiosis field and those engaged in efforts to engineer the nitrogen-fixing root nodule symbiosis in non-legume plants.

I have listed below a number of comments and suggestions which could help in improving the manuscript.

Major comments/suggestions:

1. Even in the presence of high nitrate concentrations, at or above 10 mM, wild type *L. japonicus* roots are still able to form a few underdeveloped nodules. Consequently, the nitrate inhibition of nodule initiation per se appears to be a rather ineffectual process. I think it is important to incorporate this information into their discussion and/or description of the proposed model.
2. Abstract, lines 34-36. Given that data are based on the analysis of a single species, *Lotus japonicus*, the statement seems too far reaching ("this shows that legumes...have evolved"). They could analyse selected responses in a few additional legume species to make their observations in *L. japonicus* more broadly applicable.
3. Lower KNO₃ concentrations (i.e. <<5 mM KNO₃) are not inhibitory to nodule formation. In *Arabidopsis*, expression of some IPT genes is upregulated by low but inhibited by high concentrations of nitrate or ammonia. Could thus *Lotus japonicus* lack the cytokinin biosynthesis response to high nitrogen but still maintain it at lower concentrations? In other words, have they tested the impact of lower concentration of KNO₃ on the cytokinin response (e.g. the *LjIpt3* gene expression) in *L. japonicus* roots?
4. Figure 1A. The panel seems to indicate that partially overlapping but not entirely equivalent root segments were used for the analyses. This is of concern. The susceptibility zone is defined as a root segment positioned at a fixed, proximal position to the root tip, which is not what the drawing is showing.

Minor issues:

5. Abstract, line 34. The "non-symbiotic species" term is too vague. It should be better defined what is really meant here.
6. Lines 83-84. Please provide appropriate reference to this statement.
7. Figure 1 and S1 and also other figures (e.g. Figure 6). Statistics, comparing mock to inoculated samples should be provided. I suspect that this has been omitted due to the already published data and perhaps in order to enhance clarity of the graphs. Nonetheless, these statistics are important and should be included. Also, is the iP content in mock versus inoculated roots grown in the presence of KNO₃ (panel E) significantly different?

8. Lines 91-93. "...the relative transcript abundance of Ipt2; Ipt3; Ipt4; Log1; and Log4 was lower at both one and two days post inoculation (dpi), relative to plants grown in the absence of nitrate". This is a correct statement, except for Ipt4 at 2dpi, although levels are much lower than in the mock treatment. Furthermore the steady-state level of Ipt4 mRNA is upregulated by nitrate in the mock treatment samples, which should also be discussed.
9. Line 112. "Given that iP production is inhibited by nitrate". This looks to me like attenuated rather than "inhibited". Please consider revising this and other similar statements.
10. It would be useful to include the ipt2 mutant in their analyses, as by their own work, Ipt2 is the major player in elevated cytokinin responses to rhizobial inoculation (Reid et al., 2017). I understand that the corresponding LORE1 mutant line is unavailable but was wondering whether CRISPR/Cas-based mutagenesis had been attempted?
11. Lines 134-135. The "nodule initiation" term, as used here, is incorrect. The total number of emerged, visually discernible nodules is presented and it should be referenced as such. Also, "nodule maturation" should be replaced by 'mature nodules', which is a simple and more understandable term. Furthermore, figure legends to panels B and C need to be corrected as they are switched.
12. Lines 141-142. "Under high nitrate conditions, Gifu shows a significant 35.4% reduction in ARA activity relative to nitrate free conditions (Fig 3D). Please mark this significance with an asterisk on the graph.
13. Lines 142-145. "We found ipt3-1, ipt4-1 and ipt3-2 ipt4-1 mutants all showed a significantly greater sensitivity to nitrate inhibition, exhibiting 80.2%, 64.3% and 91.7% reduction in ARA respectively relative to nitrate free conditions (Fig 3D)". This could be a direct effect of reduced nitrogenase activity, indirect effect of reduced nodule number and size, or both, which should be discussed.
14. Line 157. This should read 'nodule initiation' and not "early nodule initiation".
15. Lines 185-186. "Cytokinin application is thus able to rescue nitrate inhibition of nodule initiation, maturation and nitrogen fixation of the ipt3-2 ipt4-1 mutant". As nodule primordia were not counted in this experiment, the impact on nodule initiation cannot be judged and should not be claimed.
16. Figure 5. Description of panels B and C is switched in the figure legend. Also, use of mature (graph) versus pink (legend) nodules can be misleading. Please unify this nomenclature (e.g. mature pink nodules).
17. Figure 6. Upon M. loti inoculation, levels of RR5 and RR9 mRNAs appear significantly lower in nlp1-2 and nlp4-1 mutants as compared to wild-type. Why?
18. Line 251. Why was a higher, 10mM KNO3 concentration used in RNAseq as opposed to 5mM KNO3 used in all other experiments? Please provide a rationale for this.
19. Lines 256-257. The reference to CEP1 and CEP7 should be provided.
20. Figure S5. The legend to the figure should be more descriptive. For example, are mean values reflected by smaller dots? Also, for some data points, where transcript abundance between replicates is more similar, the mean values are not visible, which should be stated.
21. I assume that ¼ B&D used in the RNAseq experiment had no nitrogen for the first 14 days before treatment, correct? If so, this should be clearly stated.
22. The nlp1-2 appears to have diminished levels of NFR1 and NFR5 mRNAs as compared to the mock-treated wild-type control. Could the authors offer any explanation/discussion of this observation.
23. Line 291. "suppression of....., by suppressing". I am not entirely convinced that "suppression" is the best way to describe many of the observed effects. It seems that at least in several cases the response is reduced (attenuated) rather than suppressed.
24. Line 319. " may support the role of LjIPT3 as..."; should read "...LjIpt3...".
25. Lines 324-325. Miri et al., TPS (2016) 21, 178 should be cited here.
26. Line 326. " Consistent with the hyperinfection phenotype of ljlhk1,.."; should read "...Ljlhk1...".
27. Lines 327-328. "This shows that in addition to receptor signaling, cytokinin biosynthesis negatively regulates rhizobia infection." This sentence does not make sense. What is demonstrated by the hyperinfected phenotype of the ipt3-2 ipt4-1 double mutant is that de novo cytokinin biosynthesis is required to limit M. loti infection.
28. IP but not tZ levels were decreased by KNO3; should this be considered/discussed?

Reviewer #4 (Remarks to the Author):

While the phenomenon is very exciting the mechanism has not been sufficiently explored. In particular, ethylene shows strong interactions with ethylene, a possibility that should be carefully considered in this manuscript. Ethylene (ET) and Cytokinin (CK) have a well-established relationship. ie. CK induces ethylene production, that operates through a very well-studied mechanism of ACC synthase stabilization (Plant Journal 2009, 57:606 and 8 other papers cited therein).

Notably, several early studies linked nitrate suppression of nodulation with ET (Nitrate-induced ethylene biosynthesis and the control of nodulation in alfalfa. Plant Cell Environ. 1998, 21, 87–93; Nitrate inhibition of nodulation can be overcome by the ethylene biosynthesis inhibitor aminoethoxyvinylglycine. Plant Physiol. 1991, 97, 1221–1225). In addition, CK induction of ethylene has a known role in inhibition of rhizobial infection. Consequently, AVG is often used to limit ethylene inhibition of nodulation on plates, but it has not been used in this work, which means its role in the studied mechanism unclear. Ethylene is known as a major regulator of nodulation through inhibition of calcium spiking (Plant Cell. 2001, 13, 1835–1849), so could explain the effects on symbiotic signaling. Such a link would provide the paper with some insight on mechanism, which is currently lacking. Decreased sensitivity to ET could explain the increased infections in the IPT mutants seen in Fig 4B, and CK induction of ET could explain the reduced ARA in WT in Fig 5D. Either way, whether ET plays a role in CK's effect on nitrate-suppression of nodulation should be clarified.

In Fig 2 legend it is not clear what tissues are being analyzed, roots, parts of roots, nodules?

The uninoculated plants show reduced CK levels, so non-symbiotic phenotypes, particularly decrease in shoot fresh weight, given the importance of shoot-root communication in the symbiosis.

The model proposed in Fig 9 suggests that nitrate acts through NLP1/4 to inhibit CK content in three different ways, one through NF signaling, another through NSP2/ERN1, and a third, directly acting on CK content. This mish-mash of arrows doesn't match the corresponding entry in the text, which describes a linear A-> B-> C-> D pathway.

Most of the experiments use zero nitrate (not even ammonia is included) which is a suboptimal condition for nodulation and N-fixation. To determine the broader relevance of the findings, the experiments in Fig 3 must be carried out in permissive low nitrate conditions (0.5 mM KNO₃) to be convincing. See Valkov et al The functional characterization of LjNRT2.4 indicates a novel, positive role of nitrate for an efficient nodule N₂ -fixation activity. New Phytol. 2020 228:682-696.

The ipt mutants used in the study should be properly introduced in the Introduction.

It is not clear from the data presented that Lotus really responds to CK and nitrate differently than Arabidopsis, instead it likely reflects the special context of the symbiosis, an indirect effect exerted through the nod factor signaling pathway on nodule organogenesis which involves CK activation.

The Supplemental Data Set 5 should be annotated with gene functions/names as well as fold changes, means, P-values

We have revised the manuscript text according to the reviewer comments and have provided specific responses below each comment in blue. We have performed several additional experiments with our major updates listed below:

1. We performed an additional set of cytokinin measurements with the *nlp* mutants, and include this data in figure 6F. This also allowed us to update the previous figures with cytokinin measurements with increased number of biological replicates in all figures. Measurements of tZ are also now included in figure 2.
2. We performed additional qPCR to analyse expression of CYP735A in the presence of nitrate (Fig 1E) and cytokinin response regulators RR5 and RR9 (Fig 6) and NIN (FIG 8F) in the *nlp* mutants.
3. We analysed the susceptibility of cytokinin induced spontaneous nodules to nitrate inhibition in wild-type and *nlp* mutants (Fig 7F)
4. We include the proportion of mature nodules as an additional panel in all figures showing nodulation data.
5. We updated the data presentation in all figures for clarity and consistency. Where $n < 20$ we have shown all individual data points.
6. We improved the clarity of our model in figure 9.

Minor changes:

- Updated formatting to match journal requirements including removing subheadings in discussion
- Added a supplemental data file with all data used in plots

Specific responses to reviewers

Reviewer #1 (Remarks to the Author):

The manuscript reports an important finding that nitrate inhibition of nodulation occurs through regulation of cytokinin biosynthesis. The authors show that expression of cytokinin biosynthesis genes and levels of cytokinin (iP and tZ) are reduced in the root susceptible zone under high nitrate conditions. They also show that cytokinin biosynthesis mutants are hypersensitive to nitrate inhibition of nodulation and that application of cytokinins can rescue this hypersensitivity. Finally they show that Nlp1 and Nlp4 are required for the regulation of cytokinin biosynthesis genes by nitrate and that a number of nod factor signaling components are regulated by nitrate via Nlp1 and Nlp4. The paper reports a comprehensive evaluation of how cytokinin biosynthesis is regulated by nitrate during nodulation. The paper is generally well-written. I have the following comments.

1. The authors should clearly distinguish conclusions based on direct regulation of cytokinin biosynthesis by nitrate versus regulation in the presence of rhizobia. This is important since nitrate also suppresses nod factor signaling which can indirectly lead to reduced cytokinin biosynthesis. For example, the expression of *lpt4* and *Log4* and the levels of iP seem to be reduced by nitrate in mock inoculation samples (direct regulation) whereas *lpt2* and *Log1* expression is reduced by nitrate only under rhizobium inoculation.

Thank you for raising this point. We have now made this more precise in the results and discussion. For example In results lines 88-94:

“lpt2; lpt3; Log1; and Log4 was lower at both one and two days post inoculation (dpi), relative to plants grown in the absence of nitrate”

“In the absence of inoculation, only *Ipt4* and *Cyp735a* (elevated), showed significant differences in transcript levels in the nitrate condition.”

And in discussion we now highlight that the symbiotic induction is suppressed, while additional mechanisms may operate in uninoculated plants to reduce iP levels:

“While the reduction in cytokinin, in particular iP, is consistent, the mechanism underlying the response in uninoculated and inoculated roots shows some differences. In inoculated roots, where cytokinin synthesis is strongly stimulated, restriction of the symbiotic pathway impacts expression of multiple cytokinin biosynthesis genes. In uninoculated roots, we did not find significant inhibition of the same genes, despite the significant reduction in cytokinin levels. This suggests other mechanisms, such as increased cytokinin degradation, may occur in uninoculated roots exposed to high nitrate.”

2. Conclusions on regulation of nodule maturation independent of nodule initiation could be improved by evaluation what proportion of nodules were mature instead of the number of mature nodules (Lines 134-136).

We included proportion of mature nodules as an additional panel in all figures that show nodule numbers.

The figure legend in Figure 3, 5 and 7 are incorrect. Panel B shows the number of total nodules and panel C shows the number of mature nodules.

We have corrected them all.

3. Did the authors calculate ARA on a per nodule basis? Is nitrogen fixation inhibited per se by nitrate or is the reduced ARA simply due to reduced number of nodules.

We have now recalculated the data on a per nodule basis using the nodule counts for the plants used in the ARA (see below example using data from figure 3). We find this does not make a difference to our conclusions, and therefore prefer to present the data on a per plant basis as this is the way the experiment was originally performed.

4. Lines 158-159: The statement is very confusing. Revise it for clarity e.g. " *ipt3-2 ipt4-* form significantly more nodule primordia than Gifu".

We updated this sentence to read "In the absence of nitrate, *ipt3-2 ipt4-1* nodule primordia formation is not significantly different from Gifu"

5. Line 163: Consider revising "more than 3 times more"

We changed this to "significantly more"

6. Line 167 and associated text: Consider revising "nitrate resistance to nodule initiation..."

Changed to "*lpt3* and *lpt4* contribute to maintenance of nodule initiation in the presence of nitrate"

7. Line 411: Provide specifics; How many millimeters of root tip was removed?

Updated to "Root tips were removed (3mm from tip)"

Reviewer #2 (Remarks to the Author):

The manuscript submitted by Lin et al reports the role of cytokinin, which is important for regulation of nodulation, in the nitrate inhibition of nodulation. The authors took advantage of *Lotus japonicus* LORE1 insertion mutants, and examined nodulation of mutants in response to exogenous nitrate and cytokinin. Notably, they found that suppression of nodulation, caused in cytokinin biosynthesis mutants under inhibitory nitrate condition, was overcome by mutation in NLP, which is a central regulator of nitrate responses. The nitrate inhibition of nodulation was further examined in the expression of symbiosis genes that are involved in NF signaling. The authors concluded that nitrate suppresses the NF signaling, resulted in reduction of cytokinin biosynthesis, thereby attenuates nodule initiation and development.

Overall, the manuscript is comprehensive to read, the experiments are precisely described, and publications are properly cited. I have however several concerns as written below.

1. From the double mutant analysis shown in Fig. 7, it is genetically clear that *nlp1* (in the case of mature nodules) and *nlp4* are epistatic to *ipt4*, thereby CK (or cytokinin biosynthesis at least) inhibits the NLP function in suppression of nodulation by nitrate. This fits very well into the results in Figs. 3 and 5, but is totally contradictory to the title and the conclusion. I understand from Fig. 6 that NLPs are involved in the nitrate inhibition of the expression of cytokinin biosynthesis and signaling genes. Therefore, from the results shown in the manuscript two pathways should be equally considered.

Thank you for raising this, and we have tried to be more specific about these two alternatives in the discussion. The two possibilities would be that cytokinin is required for NLP function in regulation of nodule maturation compared to the alternative of CK interferes with NLP function. It is possible (or likely) that both possibilities are true due to the intricate links between cytokinin and NIN and NLP signalling. We now try to better address this second possibility.

For example we now state in discussion: "Alternatively, cytokinin may function to regulate *Nlp1* and *Nlp4* action. Cytokinin induction of the transcription factor *Nin*, which antagonises NLP function, may provide such a link" with a reference to the recent work by Nishida et al.

We disagree that the title and conclusions is contradictory to either of these possibilities, as it is clear that cytokinin can rescue some aspects of nitrate inhibition of nodulation, and we do not directly address the genetic relationship of cytokinin and NLPs in our title. We amended slightly the final conclusion to reflect that cytokinin may alter NLP function, in addition to the fact that cytokinin may mediate NLP function.

2. There can be several experiments that authors may consider in this context: One is application of CK to *nlp* mutants. If nitrate inhibits nodulation through inhibition of cytokinin biosynthesis, the effect of CK in (spontaneous) nodulation should be nitrate insensitive, contrary to the situation in WT as shown in Heckmann et al. 2011. In addition, since the gain-of-function LHK1 mutant *snf2* is nitrate sensitive (Tirichine et al. 2006), combination of *snf2* and *nlp* mutants should be nitrate insensitive as well.

Thank you for this suggestion. To test this, we performed a spontaneous nodulation assay (plants grown on BA) in WT and *nlp* mutants. Indeed *nlp1-2* is insensitive to nitrate inhibition of spontaneous nodules while *nlp4-1* retain some sensitivity. We include this as figure 7F and added associated text. This highlights that cytokinin signalling is sensitive to nitrate inhibition in addition to sensitivity of biosynthesis.

It is clear that both cytokinin biosynthesis and additional signalling components are targeted and we altered discussion to address this. "Our finding that cytokinin application can rescue some inhibitory effects of nitrate on nodule maturation and nitrogen fixation, together with the nitrate sensitivity of spontaneously formed nodules, shows that both biosynthesis and downstream cytokinin signalling are regulated by nitrate during nodule development."

3. The conclusion drawn from Fig. 8 is overstatement (description started from L. 290, L. 369, L. 379, Fig. 9, and the section title in L. 248). It is true from the results that reduction in the expression of symbiosis genes by nitrate is NLP dependent, but this degree of reduction in WT is not validated for suppression of nodulation, or for inhibition of cytokinin biosynthesis, by nitrate. Please rephrase texts appropriately. In addition, according to Fig. 8A, *Nin* is the most notable gene which expression is downregulated by nitrate. Please include *Nin* expression in WT and *nlp* mutants alongside with other symbiosis genes.

We have added *Nin* expression by qPCR in the same series, although we find a more significant impact on *Ern1* and *Nsp2*. We note that the downregulation of *Nin* shown in the heatmap is in non-symbiotic condition where *Nin* expression is already very low.

Regarding the impact on nodulation of these reductions, we now state in discussion (line 390) that the reduction in NF signalling components "alongside the reduction in symbiotically triggered cytokinin biosynthesis corresponds to a reduction in nodule development", rather than implying this connection is direct. We have also updated our figure 9 proposed model to be more precise, showing the impact of NLP dependent signalling on NF signalling and cytokinin biosynthesis.

Specific comments:

1. L. 30 "... nodule development, maturation and nitrogen fixation.", L. 31 "... nodulation and nitrogen fixation ...": To be precise, authors did not show (the activity of) nitrogen fixation per se, because plants were grown in the presence of nitrate for 3 weeks (Figs. 3 and 5). As authors mentioned in the text (L. 139), this is proxy of nodule maturation, not nitrogen fixation. Please rephrase the sentences.

In our conditions, we still observe some nitrogenase activity in the presence of nitrate, as we chose a nitrate level of 5 mM which is not completely inhibitory to the process.

While we did not directly assay nitrogen fixation, we have measured acetylene reduction in our experiments, including those with cytokinin rescue. Acetylene reduction is routinely used as a quantitative estimate of nitrogenase and nitrogen fixation activity, and we therefore believe our statements regarding nitrogen fixation are accurate.

2. L. 46 "... cytokinin synthesis and signalling.": Cytokinin transport is equally important for nodulation. Please refer Jarzyniak et al. 2021 (doi: 10.1038/s41477-021-00873-6).

We have added this reference

3. L. 54 "The AON pathway thus integrates ... soil nitrate availability.": There is no description on the AON pathway related to nitrate so far. Please describe some in the paragraph.

We added "Nitrate induction of CLE peptides allows the AON pathway to integrate..." and 2 references to the start of this sentence

4. L. 61 and thereafter as necessary: Please also refer Nishida et al. 2021 (doi: 10.1093/plcell/koab103) for NLPs in nodulation.

We added this new reference and a sentence highlighting its significance in both the introduction and discussion.

5. Figure 1: Why did authors omit CYP735A? They previously showed that the expression of CYP735A was induced after inoculation (Reid et al. 2017), and moreover, there is a marked difference in abundance between iP and tZ upon nitrate treatment. Since CYP735A is responsible for conversion of precursors of iP to those of tZ, it is expected that nitrate influences CYP735A expression. Please add the data.

We have now performed this experiment and added this data (fig 1E). Interestingly, we find Cyp735a to be induced by nitrate in uninoculated conditions, while in inoculated conditions it is not significantly changed by nitrate. This may help to explain why iP levels are reduced by nitrate, while we find no evidence for alteration of tZ by nitrate. We have added discussion on the ratio of tZ:iP changing in nitrate conditions. "specific roles for iP and tZ cytokinin types may play a role, as we only identify significant reductions in iP by nitrate, which has a net result of increasing the ratio of tZ to iP. Such a model would imply that iP is predominantly required for nodule development in the root, while tZ participates in signalling nitrate availability"

6. Figure 2 and thereafter: Why did authors omit *ipt2* mutants for analysis? They mentioned that IPT2 is "the major player in elevated cytokinin responses from the IPT family" (Reid et al. 2017). Absence of LORE1 insertion mutants (as mentioned in Reid et al. 2017) is not a rational excuse, because nowadays knockout by CRISPR is feasible even in *Lotus japonicus*. Please involve *ipt2* mutants in the analysis.

While we have highlighted *lpt2* as the major player, we continue to emphasise the genetic redundancy in cytokinin biosynthesis. We show that *lpt2* expression is inhibited by nitrate in inoculated conditions (Fig1 B) and have demonstrated a strong phenotype with the *ipt3 ipt4* double mutant in these conditions (Fig 3A and others). Inhibition of *lpt2* together with knock-out of *ipt3* and *ipt4* therefore is sufficient to produce a strong phenotype.

Producing homozygous CRISPR mutants through stable transformation of *Lotus* is not yet routine and would likely require at least a further year of time. It is unclear which additional experiments are suggested, and we believe further analyses of *ipt2* mutants would not change the overall conclusions.

7. Figure 2: The activity of IPT affects both iP and tZ contents. Please measure tZ as well.

We have now included tZ data at figures 1G and 2B. This data shows that iP is significantly reduced by nitrate in *Lotus*, while we did not detect differences in tZ between mock and nitrate treatment in wild type or any mutants analysed. This suggests iP plays the major role in sustaining nodule development, and we have added text to this effect in the results and discussion.

8. Figure 5: Application of exogenous CK reduces total and mature nodules in WT. This is fine, as authors mentioned that “excessive cytokinin accumulation can also inhibit nodulation” (L. 304). However, on the assumption that “nitrate inhibits nodule organogenesis through inhibition of cytokinin biosynthesis” as shown in the title, lower concentration of exogenous CK should be tested to know whether it is also true in WT.

We now performed this experiment and have included the data as a supplemental figure 3 and associated text. We were unable to identify a significantly increased nodulation in WT, although there is a trend for increased nodulation at 5×10^{-10} M cytokinin.

9. Figure 6: I understand that attenuation of the expression of cytokinin biosynthesis and signaling genes by nitrate requires NLP1 and NLP4, but at the same time I wonder whether it is also applicable to iP and tZ contents. This is important because some of the genes are more abundant in the mutant background, indicating higher CK accumulation. Please measure iP and tZ as shown in Fig. 1.

We have now measured cytokinin in the *nlp* mutant backgrounds. We see reduced cytokinin levels in these mutants relative to wild-type and they lose sensitivity to nitrate as expected. A recent paper by Nishida et al identifies a role for NLPs in regulation of some symbiotic genes, and we think this may explain the reduced expression of some symbiotic genes in these mutants. We have added this to the discussion along with a reference to this work.

We have also included tZ in figure 2 now, although we find no significant changes. We include the tZ levels for the *nlp* mutants as supplemental figure 4

10. Figure S3c: Some of the datapoints are not the natural numbers (i.e. 2.5 or 7.5). Are values average? If not, please revisit the data.

Thank you for picking this up, we corrected the problem. The data in this figure is now all contained in figure 7 so we have removed this supplemental figure.

Reviewer #3 (Remarks to the Author):

This is an important work that provides a mechanistic explanation for a long recognized but not well understood phenomenon of nitrate inhibition of root nodule formation in legume plants. Legumes, unlike most other plants, can tap into atmospheric nitrogen to support their growth, independent of soil nitrogen. They do so by forming root nodules that host symbiotic, nitrogen fixing bacteria. However, soil nitrogen is inhibitory and high concentrations of KNO_3 restrict nodule development. Understanding the underlying process has been a long standing goal in the effort to enhance nitrogen fixation in legumes and also to improve nitrogen uptake/use efficiency, an important agronomic trait, in non-legume crop species.

The authors now show that the main target of the nitrate-dependent regulation is the symbiosis pathway, which is attenuated by a NIN-like protein-dependent mechanism, leading to a local cytokinin deficiency that restricts nodulation. The corollary hypothesis, based on their observations, is that under nitrogen limited conditions, legumes differ from non-legumes in their cytokinin biosynthesis response to nitrate. Their results constitute a significant contribution which should be of considerable interest to a broad audience, including evolutionary and developmental biologists. The data will also be of specific interest to researchers working in the symbiosis field and those engaged in efforts to engineer the nitrogen-fixing root nodule symbiosis in non-legume plants. I have listed below a number of comments and suggestions which could help in improving the manuscript.

Major comments/suggestions:

1. Even in the presence of high nitrate concentrations, at or above 10 mM, wild type *L. japonicus* roots are still able to form a few underdeveloped nodules. Consequently, the nitrate inhibition of nodule initiation per se appears to be a rather ineffectual process. I think it is important to incorporate this information into their discussion and/or description of the proposed model.

Numerous pathways influence nitrate response. This includes both local and systemic mechanisms, and influence at multiple stages of nodule development. We added a discussion point that mechanistic insight into "how *Nlps* mediate nitrate regulation and interact with other nodule regulatory pathways to finely tune nodulation in response to the environment and plant resource requirements may allow manipulation of this process."

2. Abstract, lines 34-36. Given that data are based on the analysis of a single species, *Lotus japonicus*, the statement seems too far reaching ("this shows that legumes...have evolved"). They could analyse selected responses in a few additional legume species to make their observations in *L. japonicus* more broadly applicable.

In our abstract, we state "legumes, as exemplified by *Lotus japonicus*". We have updated the discussion to be clear that our conclusions are from *Lotus*, and that detailed analysis of cytokinin biosynthesis in other legumes, particularly where some biosynthesis genes appear to show different responses (eg soybean line 369) could be interesting studies.

3. Lower KNO₃ concentrations (i.e. <<5 mM KNO₃) are not inhibitory to nodule formation. In *Arabidopsis*, expression of some IPT genes is upregulated by low but inhibited by high concentrations of nitrate or ammonia. Could thus *Lotus japonicus* lack the cytokinin biosynthesis response to high nitrogen but still maintain it at lower concentrations? In other words, have they tested the impact of lower concentration of KNO₃ on the cytokinin response (e.g. the *LjIpt3* gene expression) in *L. japonicus* roots?

We have demonstrated in this work that in the growth system we use (plants growing on filter paper on agar slants) 5 mM nitrate has a strong inhibitory effect on nodule formation, maturation and nitrogen fixation, particularly in the *ipt* mutants. We therefore analysed this concentration.

It is possible that different concentrations, root regions or tissues may display different responses and we have tried to expand upon these possibility in the revised discussion. We moderate the language to indicate that in the conditions we tested, and particularly during nodule development a different cytokinin response is required. We also added discussion that the response may be through different mechanisms in inoculated and uninoculated plants.

4. Figure 1A. The panel seems to indicate that partially overlapping but not entirely equivalent root segments were used for the analyses. This is of concern. The susceptibility zone is defined as a root segment positioned at a fixed, proximal position to the root tip, which is not what the drawing is showing.

We have amended figure 1a to better illustrate the harvested region and adjusted the text to indicate that we analysed the root zone that *was* susceptible at the time of inoculation (at 2 dpi this is several mm proximal to the susceptible zone. This allows analysis of the region that is actively forming cell divisions and nodules at the time of harvest.

Minor issues:

5. Abstract, line 34. The “non-symbiotic species” term is too vague. It should be better defined what is really meant here.

We replace this with “species that do not form symbiotic root nodules”

6. Lanes 83-84. Please provide appropriate reference to this statement.

Here we are speculating with reference to our own work, and we have clarified this by rephrasing to “our work implies... additional mechanisms may be recruited”

7. Figure 1 and S1 and also other figures (e.g. Figure 6). Statistics, comparing mock to inoculated samples should be provided. I suspect that this has been omitted due to the already published data and perhaps in order to enhance clarity of the graphs. Nonetheless, these statistics are important and should be included. Also, is the iP content in mock versus inoculated roots grown in the presence of KNO₃ (panel E) significantly different?

We have updated figure 1 and S1 and conducted ANOVA with Tukey all-by-all comparisons to test significance of gene induction by rhizobia, as well as suppression by nitrate, and these statistics are now annotated on the figures. The iP content shown in figure 1 is significantly reduced in both mock and inoculated conditions.

8. Lines 91-93. “..the relative transcript abundance of *lpt2*; *lpt3*; *lpt4*; *Log1*; and *Log4* was lower at both one and two days post inoculation (dpi), relative to plants grown in the absence of nitrate”. This is a correct statement, except for *lpt4* at 2dpi, although levels are much lower than in the mock treatment. Furthermore the steady-state level of *lpt4* mRNA is upregulated by nitrate in the mock treatment samples, which should also be discussed.

Thank you, we corrected this error in results and now state “*lpt4* and *Cyp735a* (elevated), show significant differences...”

9. Line 112. “Given that iP production is inhibited by nitrate”. This looks to me like attenuated rather than “inhibited”. Please consider revising this and other similar statements.

We have changed these statements to “restricted” rather than inhibited

10. It would be useful to include the *ipt2* mutant in their analyses, as by their own work, *lpt2* is the major player in elevated cytokinin responses to rhizobial inoculation (Reid et al., 2017). I understand that the corresponding LORE1 mutant line is unavailable but was wondering whether CRISPR/Cas-based mutagenesis had been attempted?

We are currently establishing stable transformation and crispr, but have not yet been able to establish such lines. While we agree it is desirable, it is likely to take at least 1 year to obtain homozygous lines, and it is not clear that additional analysis of *lpt2* mutants would change our conclusions (see similar response to reviewer 2 above).

11. Lines 134-135. The “nodule initiation” term, as used here, is incorrect. The total number of emerged, visually discernible nodules is presented and it should be referenced as such. Also, “nodule maturation” should be replaced by ‘mature nodules’, which is a simple and more understandable term. Furthermore, figure legends to panels B and C need to be corrected as they are switched.

We have corrected the figure legends.

12. Lines 141-142. "Under high nitrate conditions, Gifu shows a significant 35.4% reduction in ARA activity relative to nitrate free conditions (Fig 3D). Please mark this significance with an asterisk on the graph.

Thank you for picking that up, we have corrected the figure. (P=0.02)

13. Lines 142-145. "We found ipt3-1, ipt4-1 and ipt3-2 ipt4-1 mutants all showed a significantly greater sensitivity to nitrate inhibition, exhibiting 80.2%, 64.3% and 91.7% reduction in ARA respectively relative to nitrate free conditions (Fig 3D)". This could be a direct effect of reduced nitrogenase activity, indirect effect of reduced nodule number and size, or both, which should be discussed.

This is a good point. We added some discussion about this. "Whether this reduction in nitrogen fixation is a direct effect of cytokinin, or is indirectly related to the failure of nodules to mature remains to be determined. Supporting an indirect effect is the apparent reduction in size of nodules in high nitrate conditions with the onset of fixation only occurring when nodule growth and maturation is sufficiently stimulated by cytokinin. Feedback "sanctioning" in response to reduced nitrogen fixation activity is also known to restrict nodule size (Westhoek et al. 2021). A more direct regulation of nitrogen fixation by cytokinin could then trigger such regulatory mechanisms, restricting nodule growth."

14. Line 157. This should read 'nodule initiation' and not "early nodule initiation".

corrected

15. Lines 185-186. "Cytokinin application is thus able to rescue nitrate inhibition of nodule initiation, maturation and nitrogen fixation of the ipt3-2 ipt4-1 mutant". As nodule primordia were not counted in this experiment, the impact on nodule initiation cannot be judged and should not be claimed.

We have corrected the title and sentences in this paragraph to reflect that we assessed nodule maturation and fixation.

16. Figure 5. Description of panels B and C is switched in the figure legend. Also, use of mature (graph) versus pink (legend) nodules can be misleading. Please unify this nomenclature (e.g. mature pink nodules).

We have updated all figures to read "mature nodules" and state in the text at first mention at line 140 that mature nodules "as determined by nodules acquiring a distinct pink-red colour characteristic for fully developed functional nodules expressing leghemoglobin"

17. Figure 6. Upon M. loti inoculation, levels of RR5 and RR9 mRNAs appear significantly lower in nlp1-2 and nlp4-1 mutants as compared to wild-type. Why?

This was also raised by reviewer 2, and we provide some additional context above to this point. The biosynthesis genes lpt2 and Log1, together with iP levels are also reduced in nlp mutants. Notably, NLP1 and NLP4 have significant cross signalling with symbiotic gene expression through common targets with NIN as recently demonstrated by Nishida et al (2021). We reference this and have added some discussion along these lines.

18. Line 251. Why was a higher, 10mM KNO₃ concentration used in RNAseq as opposed to 5mM KNO₃ used in all other experiments? Please provide a rationale for this.

We added the following sentence in the result section "Note that given we wished to identify primary nitrate response at early time points, we used a higher nitrate concentration (10 mM) in the RNAseq than in nodulation experiments (5 mM) where plants were exposed over a number of weeks"

19. Lines 256-257. The reference to CEP1 and CEP7 should be provided.

We added a reference to Imin et al., 2013

20. Figure S5. The legend to the figure should be more descriptive. For example, are mean values reflected by smaller dots? Also, for some data points, where transcript abundance between replicates is more similar, the mean values are not visible, which should be stated.

We updated this figure (now figure S6) as suggested, and now plot only the mean +/- SE for each gene for clearer presentation. All individual data points are available in the supplemental data file.

21. I assume that ¼ B&D used in the RNAseq experiment had no nitrogen for the first 14 days before treatment, correct? If so, this should be clearly stated.

We updated the methods section to read “14 days growth on nitrate free plates”

22. The nlp1-2 appears to have diminished levels of NFR1 and NFR5 mRNAs as compared to the mock-treated wild-type control. Could the authors offer any explanation/discussion of this observation.

Nishida 2021 recently examined NLP1, NLP4 and NIN transcriptional response, observing a similar phenomenon for some Nin target genes. It is possible NLPs play a role in regulation of symbiotic susceptibility and gene expression regulation, and we have added this to introduction and discussion (see previous comments above and to reviewer 2)

23. Line 291. “suppression of....., by suppressing” . I am not entirely convinced that “suppression” is the best way to describe many of the observed effects. It seems that at least in several cases the response is reduced (attenuated) rather than suppressed.

We have changed this in several places to reduced or restricted depending on the context

24. Line 319. “ may support the role of Ljpt3 as...”; should read “...Ljpt3...”.

corrected

25. Lines 324-325. Miri et al., TPS (2016) 21, 178 should be cited here.

We added this reference

26. Line 326. “ Consistent with the hyperinfection phenotype of ljlhk1,..”; should read “...Ljlhk1...”.

corrected

27. Lines 327-328. “This shows that in addition to receptor signaling, cytokinin biosynthesis negatively regulates rhizobia infection.” This sentence does not make sense. What is demonstrated by the hyperinfected phenotype of the ipt3-2 ipt4-1 double mutant is that de novo cytokinin biosynthesis is required to limit M. loti infection.

This sentence now reads “this shows that cytokinin biosynthesis negatively regulates rhizobia infection”

28. IP but not tZ levels were decreased by KNO3; should this be considered/discussed?

We added sentences to the discussion: “Alternatively, specific roles for iP and tZ cytokinin types may play a role, as we only identify significant reductions in iP by nitrate, which has a net result of increasing the ratio of tZ to iP. Such a model would imply that iP is predominantly required for nodule development in the root, while tZ participates in signalling nitrate availability.”

Reviewer #4 (Remarks to the Author):

While the phenomenon is very exciting the mechanism has not been sufficiently explored. In particular, ethylene shows strong interactions with ethylene, a possibility that should be carefully considered in this manuscript. Ethylene (ET) and Cytokinin (CK) have a well-established relationship. ie. CK induces ethylene production, that operates through a very well-studied mechanism of ACC synthase stabilization (Plant Journal 2009, 57:606 and 8 other papers cited therein).

Notably, several early studies linked nitrate suppression of nodulation with ET (Nitrate-induced ethylene biosynthesis and the control of nodulation in alfalfa. Plant Cell Environ. 1998, 21, 87–93; Nitrate inhibition of nodulation can be overcome by the ethylene biosynthesis inhibitor aminoethoxyvinylglycine. Plant Physiol. 1991, 97, 1221–1225). In addition, CK induction of ethylene has a known role in inhibition of rhizobial infection. Consequently, AVG is often used to limit ethylene inhibition of nodulation on plates, but it has not been used in this work, which means its role in the studied mechanism unclear. Ethylene is known as a major regulator of nodulation through inhibition of calcium spiking (Plant Cell. 2001, 13, 1835–1849), so could explain the effects

on symbiotic signaling. Such a link would provide the paper with some insight on mechanism, which is currently lacking. Decreased sensitivity to ET could explain the increased infections in the IPT mutants seen in Fig 4B, and CK induction of ET could explain the reduced ARA in WT in Fig 5D. Either way, whether ET plays a role in CK's effect on nitrate-suppression of nodulation should be clarified. Ethylene clearly plays an important part in regulation of nodulation at various stages and we agree that ethylene signalling is likely to play a role downstream of cytokinin. We have therefore added parts to the discussion to address the role of ethylene downstream of cytokinin signalling in regulating the symbiotic pathway (eg "Cytokinin shows extensive crosstalk with ethylene signalling, including through stabilisation of ACC synthase to promote biosynthesis (Hansen et al. 2009). The hyperinfection phenotypes of *Ljlhk1* (and presumably *ipt3 ipt4* mutants) are thought to stem, at least in part, from reduced stimulation of ethylene signalling (Miri et al. 2016)".

As is correctly pointed out, we do not apply AVG, because this creates an artificial environment with low ethylene signalling as well as being an auxin receptor agonist. We disagree that experimentation on cytokinin-ethylene interaction should form a part of this manuscript, which we have focussed on cytokinin biosynthesis and nitrate signalling through NLPs. We feel additional experimentation on a link with ethylene signalling would be best served by a more focused future manuscript.

In Fig 2 legend it is not clear what tissues are being analyzed, roots, parts of roots, nodules?
We have added that it is the same root tissue as shown in figure 1A

The uninoculated plants show reduced CK levels, so non-symbiotic phenotypes, particularly decrease in shoot fresh weight, given the importance of shoot-root communication in the symbiosis. It is unclear what is meant by this comment, but we have included a supplemental figure showing that the *ipt3 ipt4* double mutant has a non-symbiotic root phenotype (greater root length). We have not analysed shoot fresh weight in these mutants.

The model proposed in Fig 9 suggests that nitrate acts through NLP1/4 to inhibit CK content in three different ways, one through NF signaling, another through NSP2/ERN1, and a third, directly acting on CK content. This mish-mash of arrows doesn't match the corresponding entry in the text, which describes a linear A-> B-> C-> D pathway. We have produced a new model based on additional data included in this revision, where we show that NLPs regulate the symbiotic pathway both positively and negatively, along with the cytokinin biosynthesis. We updated the description of the model in the discussion to reflect this.

Most of the experiments use zero nitrate (not even ammonia is included) which is a suboptimal condition for nodulation and N-fixation. To determine the broader relevance of the findings, the experiments in Fig 3 must be carried out in permissive low nitrate conditions (0.5 mM KNO₃) to be convincing. See Valkov et al The functional characterization of LjNRT2.4 indicates a novel, positive role of nitrate for an efficient nodule N₂-fixation activity. *New Phytol.* 2020 228:682-696. Figure 3 is primarily concerned with the reduction in nodulation and nitrogen fixation by high (5mM) nitrate conditions. While there are a few reports of minor increases in nodulation at very low nitrate levels, nodulation is routinely evaluated in nitrate free conditions and we feel this is an appropriate control for our studies.

The *ipt* mutants used in the study should be properly introduced in the Introduction. This was an oversight which we corrected with several lines in introduction "Cytokinin biosynthesis during nodule development is controlled partially redundantly, with multiple *Ipt* and *Log* genes contributing to increased cytokinin levels. Insertion mutants in *Ljipt4* have minor phenotypes in nitrate free conditions, while *Ljipt3* knock-down or knock-out has been reported to have variable phenotypes in several studies (Chen et al. 2014; Sasaki et al. 2014; Reid et al. 2017). Synthesis of

trans-Zeatin by *LjCyp735a* is induced during nodule development, but knock-out mutants do not display a nodulation phenotype in *L. japonicus* (Reid et al. 2017).”

It is not clear from the data presented that Lotus really responds to CK and nitrate differently than Arabidopsis, instead it likely reflects the special context of the symbiosis, an indirect effect exerted through the nod factor signaling pathway on nodule organogenesis which involves CK activation. We have demonstrated that cytokinin levels in both inoculated and uninoculated conditions are restricted in Lotus, which differs to report from other species, including Arabidopsis. We agree that in the context of symbiosis, this likely reflects an inhibition of the symbiotic signalling pathway activation of cytokinin biosynthesis. We expanded discussion on this topic to address both the symbiotic and non-symbiotic condition.

The Supplemental Data Set 5 should be annotated with gene functions/names as well as fold changes, means, P-values

We have updated this data set to include the gene annotations and provide an additional supplemental data set with differential expression statistics (log₂ fold-change, raw and adjusted P-values) as suggested.

REVIEWERS' COMMENTS

Reviewer #1 (Remarks to the Author):

The revised version has satisfactorily addressed the concerns raised previously.

Reviewer #2 (Remarks to the Author):

In this revised manuscript, the authors sufficiently addressed my concerns.

Reviewer #3 (Remarks to the Author):

The revised manuscript has been significantly improved and most of my previous comments/suggestions have been satisfactorily addressed. The writing, however, requires additional editorial work as it is sloppy in places (e.g. switching unnecessarily between past and present tenses or missing necessary punctuation).

Additional, minor comments:

1. I could not detect any changes to Figure 1a. Is this a simple omission?
2. Lines 144-145: Equating pink, mature looking nodules with functional nodules is somewhat problematic. This is highlighted by a disproportionately higher impact of nitrate on the activity of nitrogenase versus the proportion of mature looking nodules. A more careful statement should be offered.
3. Line 193: "an insignificant increase in nodule numbers" is an incorrect statement. There is no such a thing. Please consider rephrasing this.
4. Lines 268-271: The primary response to nitrate is rapid and is also known to occur at micro-molar concentrations. Therefore, their explanation for using a higher, 10 mM nitrate concentration in the RNAseq experiments is questionable. Perhaps something simple would suffice, such as pointing to the fact that unlike lower nitrate concentrations, 10 mM is quite restrictive to nodule formation, hence it was used in the RNAseq experiments.
5. Line 335. "under restrictive nitrate conditions" is an imprecise statement; under nitrate supply conditions that restrict nodulation....
6. Lines 355-356: "This shows that cytokinin biosynthesis negatively regulates rhizobia infection..." This is an incorrect statement. It should read: This shows that de-novo cytokinin biosynthesis is required for negative regulation of rhizobial infection.
7. Line 373: "This reduction was evident in both reduced expression analysis of biosynthesis genes...". This is an awkward sentence; please rephrase it.
8. Lines 404: "MtNlp1 and LjNlp4 are able to interact and/or compete with Nin to block Nin function.." and sentences that follow:. As proteins are being referred to, the corresponding symbols should not be italicized.

Reviewer #4 (Remarks to the Author):

The experiments in the paper "Nitrate restricts nodule organogenesis through inhibition of cytokinin biosynthesis in *Lotus japonicus*" are nicely designed and explained with clear results. However, the paper is not convincing in terms of showing a direct role for CK signaling in the nitrate response, instead the authors show that suppression of nodulation by nitrate results in suppression of CK signaling/biosynthesis. The action of CK downstream of NF signaling (including

the induction of NIN by CK) was reported about 15 years ago. So, it has been known for a long time that anything that affects NF signaling will of course affect CK biosynthesis/signaling (along with every other process involved in nodule formation, NIN-expression, nodule development, N-fixation etc.).

The promise of this paper was a novel direct connection between nitrate signaling and CK, with the added aspect that this feature was special to legumes. In my opinion, neither of these things has been achieved, the control showed of nitrate signaling over CK is indirect, and its not clear that legume responses to nitrate with regards to CK are that different.

In the abstract, the following statements by the authors were made:

"Low nitrate conditions provide a permissive state for induction of cytokinin by symbiotic signalling"

"...high nitrate is inhibitory to cytokinin accumulation and nodule establishment in the root zone susceptible to nodule formation."

These statements are true, but do not represent an advance in our understanding of CK role in nodulation. Of course, low nitrate conditions are permissive for nodulation and so are permissive for CK signaling/biosynthesis. High nitrate blocks nodule formation, so it follows that it blocks the CK signaling/biosynthesis that occurs during nodule formation. The converse is also obvious. The current state of knowledge was stated by reviewer 1: "...nitrate also suppresses nod factor signaling which can indirectly lead to reduced cytokinin biosynthesis."

Another statement in the abstract is:

"This reduction of symbiotic cytokinin accumulation was further exacerbated in cytokinin biosynthesis mutants, which display hypersensitivity to nitrate inhibition of nodule development, maturation and nitrogen fixation."

Of course, we expect less CK to accumulate in CK biosynthesis mutants. And we also expect such mutants to form smaller less well-developed nodules which would fix less N, this has already been well described in other studies. The hypersensitivity, which suggests that CK antagonizes N-suppression of nodulation is the most interesting finding in this study, but the mechanism is not explored. This would be interesting if it could be linked to changes in regulators such as CEPs, CLEs, ethylene or GA.

Another statement from the abstract:

"These inhibitory impacts of nitrate on symbiosis occur in a Nlp1 and Nlp4 dependent manner and contrast with the positive influence of nitrate on cytokinin biosynthesis that occurs in species that do not form symbiotic root nodules. Altogether this shows that legumes, as exemplified by *Lotus japonicus*, have evolved a different cytokinin response to nitrate compared to non-legumes.

Since we know that NLP1/4 control nitrate suppression of nodulation, they presumably also control nitrate suppression of the CK signaling/biosynthesis that occurs during nodulation. The authors conclusions that "...NLPs regulate the symbiotic pathway both positively and negatively, along with the cytokinin biosynthesis." is hardly surprising or interesting since previous publications show they both regulate nodule formation. Any 'special' interaction between nitrate and CK signaling can be attributed to the fact that NF-signaling, which of course is only present in legumes (and nodulating non-legumes) is inhibited by nitrate. The decrease in root CK levels observed under non-symbiotic conditions is interesting, but since it wasn't accompanied by decreased expression of CK biosynthesis genes may be a consequence of increased root-to-shoot translocation of CK (a possibility mentioned by the authors), which is well documented in other plants and which would be expected to vary across experimental conditions and so does not necessarily suggest differences in the relationship between nitrate and CK between legumes and other species.

Overall, the study is solid, but does not greatly advance our knowledge.

The model still draws direct arrows from CK to all points (NF perception, NSP2/ERN1, and CK) but it seems more likely that effects on NSP2/ERN1/CK all stem from blocking NF-signaling which controls all the rest.

There are parts missing in the legend in Fig 7 (F missing, E needs brackets).

We have addressed the reviewer's comments in blue below

REVIEWERS' COMMENTS

Reviewer #1 (Remarks to the Author):

The revised version has satisfactorily addressed the concerns raised previously.

Reviewer #2 (Remarks to the Author):

In this revised manuscript, the authors sufficiently addressed my concerns.

Reviewer #3 (Remarks to the Author):

The revised manuscript has been significantly improved and most of my previous comments/suggestions have been satisfactorily addressed. The writing, however, requires additional editorial work as it is sloppy in places (e.g. switching unnecessarily between past and present tenses or missing necessary punctuation).

We have read through the manuscript again and made minor edits to ensure punctuation and writing is as clear as possible.

Additional, minor comments:

1. I could not detect any changes to Figure 1a. Is this a simple omission?

We updated figure 1a with wording "harvested region" and by changing the associated text to describe the region analysed.

2. Lines 144-145: Equating pink, mature looking nodules with functional nodules is somewhat problematic. This is highlighted by a disproportionately higher impact of nitrate on the activity of nitrogenase versus the proportion of mature looking nodules. A more careful statement should be offered.

We removed the word "functional", and maintain the description as "fully developed nodules". We have used the wording "nodule maturation" here to describe pink nodules and in the following paragraph separately describe the impact on nitrogen fixation (through acetylene reduction) which can have distinct regulation.

3. Line 193: "an insignificant increase in nodule numbers" is an incorrect statement. There is no such a thing. Please consider rephrasing this.

Thank you, we have fixed this to read "did not significantly impact nodule numbers"

4. Lines 268-271: The primary response to nitrate is rapid and is also known to occur at micro-molar concentrations. Therefore, their explanation for using a higher, 10 mM nitrate concentration in the RNAseq experiments is questionable. Perhaps something simple would suffice, such as pointing to the fact that unlike lower nitrate concentrations, 10 mM is quite restrictive to nodule formation, hence it was used in the RNAseq experiments.

We simplified the description to read "we used a higher nitrate concentration (10 mM) in the RNAseq to ensure a robust inhibition of nodulation signalling"

5. Line 335. “under restrictive nitrate conditions” is an imprecise statement; under nitrate supply conditions that restrict nodulation....

We replaced this with “under nitrate conditions that restrict nodulation”

6. Lines 355-356: “This shows that cytokinin biosynthesis negatively regulates rhizobia infection...” This is an incorrect statement. It should read: This shows that de-novo cytokinin biosynthesis is required for negative regulation of rhizobial infection.

We changed this sentence as suggested.

7. Line 373: “This reduction was evident in both reduced expression analysis of biosynthesis genes...”. This is an awkward sentence; please rephrase it.

We removed the word “analysis” so the sentence now reads “This reduction was evident in both reduced expression of biosynthesis genes...”

8. Lines 404: “MtNlp1 and LjNlp4 are able to interact and/or compete with Nin to block Nin function...” and sentences that follow: As proteins are being referred to, the corresponding symbols should not be italicized.

We corrected this

Reviewer #4 (Remarks to the Author):

The experiments in the paper “Nitrate restricts nodule organogenesis through inhibition of cytokinin biosynthesis in *Lotus japonicus*” are nicely designed and explained with clear results. However, the paper is not convincing in terms of showing a direct role for CK signaling in the nitrate response, instead the authors show that suppression of nodulation by nitrate results in suppression of CK signaling/biosynthesis. The action of CK downstream of NF signaling (including the induction of NIN by CK) was reported about 15 years ago. So, it has been known for a long time that anything that affects NF signaling will of course affect CK biosynthesis/signaling (along with every other process involved in nodule formation, NIN-expression, nodule development, N-fixation etc.).

The promise of this paper was a novel direct connection between nitrate signaling and CK, with the added aspect that this feature was special to legumes. In my opinion, neither of these things has been achieved, the control showed of nitrate signaling over CK is indirect, and its not clear that legume responses to nitrate with regards to CK are that different.

In the abstract, the following statements by the authors were made:

“Low nitrate conditions provide a permissive state for induction of cytokinin by symbiotic signalling”
“...high nitrate is inhibitory to cytokinin accumulation and nodule establishment in the root zone susceptible to nodule formation.”

These statements are true, but do not represent an advance in our understanding of CK role in nodulation. Of course, low nitrate conditions are permissive for nodulation and so are permissive for CK signaling/biosynthesis. High nitrate blocks nodule formation, so it follows that it blocks the CK

signaling/biosynthesis that occurs during nodule formation. The converse is also obvious. The current state of knowledge was stated by reviewer 1: "...nitrate also suppresses nod factor signaling which can indirectly lead to reduced cytokinin biosynthesis."

Another statement in the abstract is:

"This reduction of symbiotic cytokinin accumulation was further exacerbated in cytokinin biosynthesis mutants, which display hypersensitivity to nitrate inhibition of nodule development, maturation and nitrogen fixation."

Of course, we expect less CK to accumulate in CK biosynthesis mutants. And we also expect such mutants to form smaller less well-developed nodules which would fix less N, this has already been well described in other studies. The hypersensitivity, which suggests that CK antagonizes N-suppression of nodulation is the most interesting finding in this study, but the mechanism is not explored. This would be interesting if it could be linked to changes in regulators such as CEPs, CLEs, ethylene or GA.

Another statement from the abstract:

"These inhibitory impacts of nitrate on symbiosis occur in a Nlp1 and Nlp4 dependent manner and contrast with the positive influence of nitrate on cytokinin biosynthesis that occurs in species that do not form symbiotic root nodules. Altogether this shows that legumes, as exemplified by *Lotus japonicus*, have evolved a different cytokinin response to nitrate compared to non-legumes."

Since we know that NLP1/4 control nitrate suppression of nodulation, they presumably also control nitrate suppression of the CK signaling/biosynthesis that occurs during nodulation. The authors conclusions that "...NLPs regulate the symbiotic pathway both positively and negatively, along with the cytokinin biosynthesis." is hardly surprising or interesting since previous publications show they both regulate nodule formation. Any 'special' interaction between nitrate and CK signaling can be attributed to the fact that NF-signaling, which of course is only present in legumes (and nodulating non-legumes) is inhibited by nitrate. The decrease in root CK levels observed under non-symbiotic conditions is interesting, but since it wasn't accompanied by decreased expression of CK biosynthesis genes may be a consequence of increased root-to-shoot translocation of CK (a possibility mentioned by the authors), which is well documented in other plants and which would be expected to vary across experimental conditions and so does not necessarily suggest differences in the relationship between nitrate and CK between legumes and other species.

Overall, the study is solid, but does not greatly advance our knowledge.

We agree with the reviewer that the role of cytokinin in nodulation has been well established. We believe the novelty of our study comes from establishing that nitrate inhibits cytokinin biosynthesis during this process, which differs from the induction of cytokinin by nitrate in non-nodulating species.

We have included additional text in the discussion to highlight the point raised that effects of nitrate on cytokinin biosynthesis may also be a general consequence of the effect of nitrate on nodulation signalling. In particular, we included in the discussion "The inhibition of cytokinin biosynthesis genes we detected occurs in genes that are upregulated during symbiosis. This indicates that the reduction is likely, at least in part, to reflect the impact of nitrate on the nodulation signalling pathway. However, the reduced cytokinin levels detected in both inoculated and uninoculated plants supports the view that nitrate restriction of cytokinin biosynthesis in *L. japonicus* is enacted through both nodule-dependent and nodule-independent signalling processes."

The model still draws direct arrows from CK to all points (NF perception, NSP2/ERN1, and CK) but it seems more likely that effects on NSP2/ERN1/CK all stem from blocking NF-signaling which controls all the rest.

We updated the model for the last submission to remove some arrows and to reflect that biosynthesis is controlled by both the symbiotic pathway (induction by inoculation) and nitrate directly (as evident by reduced CK levels in both inoculated and uninoculated plants).

There are parts missing in the legend in Fig 7 (F missing, E needs brackets).

We have added a description of F to the figure legend and updated the formatting.